



# Dynamic crack propagation in weak snowpack layers: Insights from high-resolution, high-speed photography

Bastian Bergfeld[1], Alec van Herwijnen[1], Benjamin Reuter[2], Grégoire Bobillier[1], Jürg Dual[3] and Jürg Schweizer[1]

[1] WSL Institute for Snow and Avalanche Research SLF, Davos, Switzerland
[2] Météo-France, CNRS, CNRM, Centre d'Etudes de la Neige, Grenoble, France
[3] Institute for Mechanical Systems, ETH Zurich, Zurich, Switzerland

*Correspondence to*: Bastian Bergfeld (bergfeld@slf.ch)

**Abstract.** To assess snow avalanche release probability, information on failure initiation and crack propagation in weak
snowpack layers underlying cohesive slab layers are required. With the introduction of the Propagation Saw Test (PST) in the
mid-2000s, various studies used particle tracking analysis of high-speed video recordings of PST experiments to gain insight
into crack propagation processes, including slab bending, weak layer collapse, crack propagation speed and the frictional
behavior after weak layer fracture. However, the resolution of the videos and the methodology used did not allow insight into
dynamic processes such as the evolution of crack speed within a PST or the touchdown distance, which is the length from the
crack tip to the trailing point where the slab sits on the crushed weak layer at rest again. Therefore, to study the dynamics of
crack propagation we recorded PST experiments using a powerful portable high-speed camera with a horizontal resolution of
1280 pixels at rates up to 20,000 frames per second. By applying a high-density speckling pattern on the entire PST column,
we then used digital image correlation (DIC) to derive high-resolution displacement and strain fields in the slab, weak layer,
and substrate. The high frame rates allowed time derivatives to obtain velocity and acceleration fields. On the one hand, we
demonstrate the versatile capabilities and accuracy of the DIC method by showing three PST experiments resulting in slab
fracture, crack arrest and full propagation. On the other hand, we present a methodology to determine relevant characteristics
of crack propagation: the crack speed and touchdown distance within a PST, and the specific fracture energy of the weak layer.
To estimate the effective elastic modulus of the slab and weak layer as well as the weak layer specific fracture energy we used
a recently proposed mechanical model. A comparison to already established methods showed good agreement. Furthermore,
our methodology also provides insight into the three different propagation results found with the PST and reveals intricate
dynamics that are otherwise not accessible.

## 1 Introduction

Snow avalanches range among the most prominent natural hazards that threaten infrastructure and people in mountain regions.
While avalanches come in many different types and sizes, here we focus on dry-snow slab avalanches, as these are typically
the most dangerous (McClung and Schaerer, 2006). Dry-snow slab avalanche release is the result of a sequence of fracture
processes. Failure initiation induced by external loading or the coalescence of sub-critical failures can lead to a localized crack
of critical size so that rapid crack propagation starts (onset of crack propagation) and the slab-weak layer system becomes
unstable. In the subsequent dynamic crack propagation phase, the crack self-propagates across the slope without the need of
additional load beside the load inserted by the slab. Avalanche release then occurs if the gravitational pull on the slab
overcomes frictional resistance to sliding and to cause cracks at the crown, flank and stauchwall of the forming avalanche
(Schweizer et al., 2003).

While avalanche release is a large scale process (slope scale, up to several hundreds of meters), the process zones of the
preceding fractures occur on much smaller scale (snowpack scale, centimeters to decimeters (Sigrist et al., 2005)). At the small
scale, the snowpack consists of layers with peculiar mechanical properties due to their complex microstructure, which is often



anisotropic (Walters and Adams, 2014). Studying fracture processes related to avalanching thus requires experiments that are large enough to map the overall slope scale processes, but also detailed enough to resolve the driving processes at the snowpack scale.

Quantities of interest during self-sustained crack propagation are the speed of the propagating crack, the touchdown distance, which is the length from the crack tip to the trailing point where the slab comes into contact with the crushed weak layer, and

the specific fracture energy of the weak layer (e.g., van Herwijnen et al., 2010;Schweizer et al., 2011;van Herwijnen et al., 2016b)

Despite the importance of crack speed on the crack propagation process (Gross and Seelig, 2001), our knowledge on crack propagation speed is rather limited, in particular over large distances. High-speed photography of Propagation Saw Tests, a fracture mechanical field experiment for snow (Sigrist et al., 2006;Gauthier and Jamieson, 2006a), combined with Particle

Tracking Velocimetry (PTV) provided new insight into weak layer fracture and crack propagation (Schweizer et al., 2011;van Herwijnen et al., 2016b;van Herwijnen et al., 2016a;van Herwijnen et al., 2010). Results showed a consecutive settlement of the slab during weak layer fracture and compaction (van Herwijnen and Jamieson, 2005;van Herwijnen and Heierli, 2010). Crack propagation speeds derived using threshold values for slope normal displacement range from 10 to 50 ms$^{-1}$ (van Herwijnen and Birkeland, 2014;van Herwijnen and Jamieson, 2005;van Herwijnen et al., 2016b;Bair et al., 2014). Therefore,

speeds are in line with theoretical predictions of incipient (McClung, 2005) and asymptotic speeds (Heierli et al., 2008b). Still, an inter comparison to an alternative experimental technique deriving crack speeds in PSTs is missing, but needed since the current methodology is based on the assumption that collapse is in line with crack advance. An increase of crack speed with increasing slab density and increasing collapse height was observed (van Herwijnen and Jamieson, 2005;van Herwijnen and Birkeland, 2014). For the touchdown distance, experimental data are very limited. Solely Bair et al. (2014) reported

experimentally estimated touchdown distances that ranged from 2.5 to 3.3 m and were therefore twice a large as what was predicted with a model suggested by Heierli (2008). With regard to weak layer specific fracture energy $w_f$, different methodologies exist yielding different results. Sigrist and Schweizer (2007) were the first to estimate $w_f$ (mean 0.07 Jm$^{-2}$) with field experiments in combination with finite element (FE) modelling. Their method requires a Propagation Saw Test to determine the critical cut length and a snow micro-penetrometer measurement (SMP; Schneebeli and Johnson, 1998) to

estimate the effective elastic modulus of the slab. However, Schweizer et al. (2011) used the same approach but reported much larger values, typically around 1 Jm$^{-2}$. This discrepancy was in part attributed to a different signal processing method for analyzing SMP force signals (Marshall and Johnson, 2009;Löwe and van Herwijnen, 2012) resulting in lower estimates of the slab effective modulus. This emphasizes the strong dependence of specific fracture energy on the elastic modulus, which comes along with this method. To resolve this discrepancy, van Herwijnen et al. (2016b) presented a field-based experimental

approach to determine the effective elastic modulus and the specific fracture energy at the same time. They found values for the specific fracture energy ranging from 0.08 to 2.7 Jm$^{-2}$. Similar values (0.5 and 2 Jm$^{-2}$) were also found by integrating the penetration force signal of the SMP over the weak layer thickness (Reuter et al., 2019;Reuter et al., 2015). Hence, there is no single method available to derive consistent values of the important metrics describing crack propagation.

The aim of the present work is to introduce a field applicable methodology that brings insight into the dynamics of crack

propagation and in particular allows deriving characteristic measures of crack propagation such as speed, touchdown distance and specific fracture energy. To this end, we employed a portable high-speed camera (up to 20,000 frames per second) and recorded densely speckled flanks (or side walls) of PST experiments. These sequences of images were then used to perform digital image correlation (DIC), which provide full-field displacement and strain fields. We show exemplary results of three flat field PST experiments that resulted in slab fracture (SF), crack arrest (ARR) or crack propagation until the far end of the

beam (END). For the latter, we also calculated the evolution of the crack speed along the PST beam from the displacement of the slab as well as from alternative methods based on the acceleration of the slab or the strain of the weak layer. From the



downward velocity field of the slab we estimated the touchdown distance. Finally, we computed weak layer specific fracture energy as well as weak layer and slab elastic modulus from the displacement field of the slab.

## 2. Methods

We performed fracture mechanical field experiments based on the Propagation Saw Test (PST) design on three measurements days on a flat and uniform site close to Davos, Switzerland. Using high-speed videos of the experiments, we applied digital image correlation (DIC) to derive high-resolution displacement and strain fields of the slab, weak layer and substrate.

### 2.1. Field measurements

On each of the three measurement days, we performed a PST. This is a standard fracture mechanical test for snow (Sigrist and
Schweizer, 2007;Gauthier and Jamieson, 2006b) where a 30 cm wide column is isolated and an artificial cut is introduced within a weak snow layer until, at the critical cut length $r_c$, a self-propagating crack starts. Unlike the standard guidelines (Greene et al., 2016) for a PST, which recommend a beam length of 120 cm, our PSTs were at least 230 cm long.

Close to the PST experiment, we characterized the snowpack with a traditional manual snow profile following Fierz et al. (2009). Density was measured using a 100 $cm^3$ cylindrical density cutter (38 mm diameter) with a vertical resolution of 5 cm.
Spatial variability of the snowpack along the PST beams was assessed with snow micro-penetrometer (SMP) measurements approximately every 50 cm along the PST experiments.

The exposed side wall of the PST was speckled with black ink (Indian Ink, Lefranc & Bourgeois) applied with a commercial garden pump sprayer. Using a high-speed camera (Phantom, VEO710), we filmed the entire speckled wall of the PST experiment with rates up to 20,000 frames per second and a horizontal resolution of 1280 pixel. We adjusted the vertical
resolution for each PST individually to maximize frame rate and recording duration. Limited by the 18 GB internal memory of the camera, the actual image resolution and frame speed determine the duration of the recorded movie, which did not always allow to film the full sawing phase prior to crack propagation in the PST experiment. We attached a circular marker to the tip of the 2 mm thick snow saw to determine the location of the saw tip in all frames using particle tracking.

To avoid perspective distortion of the speckled PST wall, we aligned the camera vertically and horizontally perpendicular to
the wall and aimed the optical axis of the camera at the center of the PST, both horizontally and vertically.

### 2.2. Image processing

**Camera distortion correction**

To correct for radial and tangential image distortion introduced by the camera lens we used calibration factors using a pinhole camera model. The distortion coefficients $k_1$, $k_2$, $k_3$, $p_1$, $p_2$ as well as the camera model were estimated using chessboard
calibration images and routines using OpenCV for Python (Bradski and Kaehler, 2008). The distortion coefficients are constant as these only depend on the lens characteristics.

In the field, we changed the image resolution to allow for longer recording times and higher frame speeds. We thus scaled the camera model matrix accordingly. Camera calibration was performed on the fly while correlating the images with DICengine (Turner, 2015). As a result, black curved boarders are introduced in the corrected images (Figure 1)

**Digital image correlation**

For the digital image correlation (DIC) analysis we used DICengine, an open source software provided by Sandia National Laboratories (Turner, 2015). In the images, a region of interest (ROI) was selected encompassing the speckled PST wall. To derive displacement and strain fields of the wall, the ROI was further subdivided into quadratic DIC-subsets with a certain side length and step size (Table 1). The position and deformation of each DIC-subset was then tracked over time. The first



frame of the movie was used as reference image. In order to find the unique DIC-subsets in all subsequent frames, the DIC-subsets were allowed to translate, rotate and deform in normal and shear. For an arbitrary DIC-subset $i$, we thus obtained the initial horizontal and vertical position $(x_i, z_i)$ within the reference image as well as the time dependent horizontal and vertical displacement $u_i(t)$, $w_i(t)$ relative to the initial position, the rotation, normal strain $\varepsilon_{zz,i}(t)$ and shear strain $\varepsilon_{xz,i}(t)$.

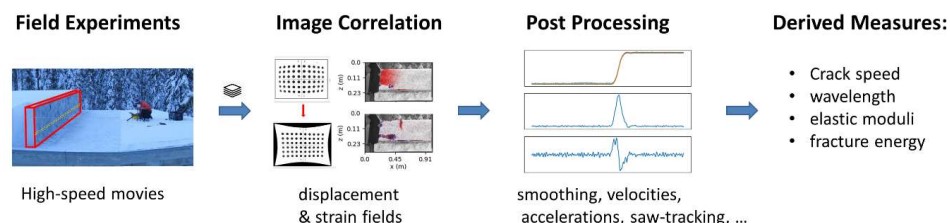

**Figure 1: Schematic drawing of data processing. After the field experiments were filmed, the frames were analyzed using digital image correlation. In a further step, the displacements obtained with DIC were smoothed and fields of velocity and acceleration were derived before quantities relevant for crack propagation were estimated.**

**Post processing**

After correlation, we used Python 3.7 to post process the DIC output. Since the obtained measures are in image space, we first
converted them to real space with meter units. For this, we manually picked a reference length in an image and determined the conversion factor. We also changed the origin and orientation of the coordinate system to the upper left corner of the slab with $x$ positive right and $z$ positive downwards.

In the next step, we divided the ROI into three regions: (i) slab, (ii) weak layer and (iii) substrate. This was done by drawing an upper and lower boundary of the weak layer manually into the displacement field after fracture. All DIC-subsets above the
upper boundary were assigned to the slab, all those below the lower boundary to the substrate and all in between to the weak layer.

Since the frame rate of the recorded videos is high compared to the typical time of crack propagation, we smoothed the displacements $u_i(t)$, $w_i(t)$. The smoothing was done using a 3rd order Savitzky-Golay filter (Savitzky and Golay, 1964) with a window size of 201 frames. To compute velocity and acceleration of the DIC-subsets, the first and second derivative of the
smoothed displacement curves were taken (Figure 1).

**Tracking the saw**

The current cut length, while sawing into the weak layer, is a crucial parameter to model the resulting slab deformation. Therefore, we tracked the dot mounted on the tip of the saw along the movie frames using DICengine's tracking functionality (Turner, 2015). Since the dot had a certain distance from the PST sidewall, there was an offset $r_{\text{off}}$ between the coordinates of
the dot $r_{\text{dot}}$ and the actual cut length $r = r_{\text{dot}} - r_{\text{off}}$ (Figure 2).





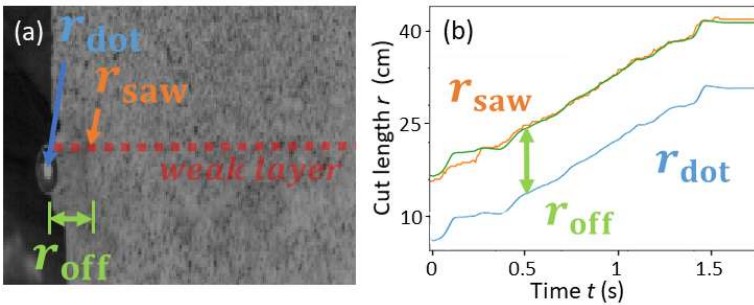

**Figure 2: (a) Close up of the saw end of a PST experiment. The saw has cut the weak layer (dashed red line) already up to the orange arrow, whereas the camera perspective causes the tracked dot (blue arrow) appearing with an offset ($r_{off}$) to the current position ($r_{saw}$). (b) Offset correction of the automatic detected saw location (blue) to the corrected cut length (green).**

### 2.3. Mechanical properties

To determine mechanical properties, specifically the effective elastic modulus of the slab $E_{sl}$ and the weak layer specific fracture energy $w_f$, a mechanical model is required to fit the experimental data (e.g. van Herwijnen et al. (2016b)). Deriving mechanical properties was done using two different approaches based on the displacement field and one approach based on the SMP measurements.

First, we followed the methodology described by van Herwijnen et al. (2016b) which is based on fitting the formulation of

mechanical energy provided by Heierli et al. (2008a). We will call this the "VH" method. van Herwijnen et al. (2016b) estimated the effective elastic modulus $E_{sl}^{VH}$ of the slab and the weak layer specific fracture energy $w_f^{VH}$ from the change in mechanical energy $V_m(r)$ with cut length $r$. Using the theorem of Clapeyron, the mechanical energy $V_m(r) = -\frac{1}{2}V_p(r)$ of the slab is connected to the loss in gravitational potential energy $V_p$, which is computed from slab displacements and for different cut lengths $r$ in the PST. Fitting the expression for the mechanical energy $V_m\left(E_{sl}^{VH}, \nu, D, \rho, \theta, r\right)$ (Heierli et al., 2008a; eqs 1 &

5) with the elastic modulus of $E_{sl}^{VH}$ as a free parameter (Figure 3a). Here, $\nu_{sl}$ is the Poisson´s ratio (assumed to be 0.25) , $D$ is slab thickness, $\rho$ is mean slab density and $\theta$ is the slope angle. The specific fracture energy of the weak layer is then obtained by numerical differentiation of the mechanical energy $w_f^{VH} = -\frac{d}{dr}V_m\Big|_{r=r_c}$ at the critical cut length $r_c$.





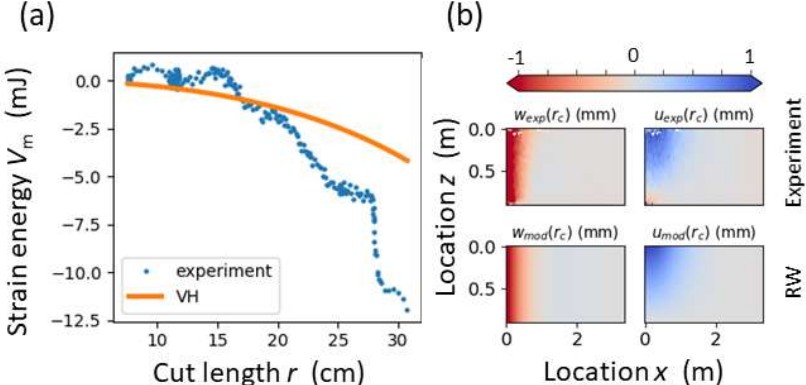

**Figure 3: (a) Strain energy $V_m(r)$ stored in the slab is decreasing with cut length; the orange line is a fit of the theoretical expression of $w_f^{VH}$ with slab elastic modulus as a free parameter. (b) vertical ($w$) and horizontal ($u$) displacement fields at $r = r_c$ in the left and right column, respectively. The top figures show the measured displacement fields while the bottom figures are the respective modelled ones using the approach described by Rosendahl and Weissgraeber (2020).**

As an alternative approach, we used the model suggested by Rosendahl and Weissgraeber (2020), which we will call the "RW" method. Their model, consisting of a Timoshenko beam sitting on a weak layer represented by smeared springs, predicts horizontal (along the PST beam length $l$, $x$-direction) and vertical (along $D$, $z$-direction) slab displacements $u(x, z)$ and $w(x, z)$ for different cut lengths $r$. Required model parameters are geometrical PST parameters ($D$, $l$, $\theta$, beam width $b$ and weak layer thickness $d$) and snow mechanical parameters ($E_{sl}^{RW}, E_{wl}^{RW}$: elastic modulus of the slab and weak layer, $\nu_{sl}, \nu_{wl}$: Poisson's ratios of the slab and the weak layer, both assumed to be 0.25). To derive the elastic modulus of the slab $E_{sl}^{RW}$ and the weak layer $E_{wl}^{RW}$ we computed a residual between the measured ($u_{exp}, w_{exp}$) and modelled ($u_{RW}, w_{RW}$) displacements (cf. Figure 3b, top and bottom line, respectively):

$$Residual(E_{sl}^{RW}, E_{wl}^{RW}) = \sum_{k=0}^{SS} \left| u_{RW}^k(E_{sl}^{RW}, E_{wl}^{RW}) - u_{exp}^k \right| + \left| w_{RW}^k(E_{sl}^{RW}, E_{wl}^{RW}) - w_{exp}^k \right|,$$

where the sum is over all DIC-subsets SS contained in the slab. Then, we used the least squares optimization routine from scipy (Virtanen et al., 2020) to find the local minimum of the residual, namely the optimal set of $E_{sl}^{RW}$ and $E_{wl}^{RW}$. The weak layer fracture energy $w_f^{RW}$ is obtained by $w_f^{RW} = G_I + G_{II}$ where $G_I = \frac{E_{wl}^{RW}}{2t} w_{RW}(r = r_c)^2$ and $G_{II} = \frac{E_{wl}^{RW}}{2t(\nu_{wl}-1)} u_{RW}(r = r_c)^2$ are the contributions from mode I and mode II, respectively.

As a third approach we used the SMP measurements. Effective elastic modulus $E_{sl}^{BR}$ and weak layer specific fracture energy $w_f^{BR}$ were derived from the SMP signals as described by Reuter and Schweizer (2018) following the approach for signal interpretation suggested by Löwe and van Herwijnen (2012).

**2.4. Crack propagation properties**

As characteristics of crack propagation we compute the temporal evolution of crack speed and touchdown distance for PST experiment #3 (PST3) where the crack travelled to the very end of the beam.

**Crack speed**

We investigated three methods to derive crack speed. First, we used a similar method as described by van Herwijnen and Jamieson (2005). As the crack propagates through the weak layer, the weak layer collapses, and displacement curves of DIC-subsets in the slab gradually show settlement (Figure 4a). The time delay $\Delta t$ between the displacement curves is then used to calculate crack speed. Therefore, the time when $w(t)$ of a DIC-subset exceeds the threshold value (0.2 mm, typically the

standard deviation of $w$) is noted and linked with the $x$-position of the given DIC-subset (Figure 4b, bottom left, blue dots). We then computed the median for all DIC-subsets with the same $x$-location. (Figure 4, bottom graphs, orange pluses). As crack speed $c = \Delta x / \Delta t$, the crack speed evolution is derived as the slope of linear fits to overlapping moving windows (Figure 4b and c, bottom graphs, green background). Uncertainty in speed is assessed using the 95% confidence interval of the fit (Figure 4,

bottom right, blue region). As this method is based on the displacement we will indicate this with a superscript "disp" in the speed estimates $c^{\text{disp}}$. In contrast to our approach, van Herwijnen and Jamieson (2005) used several different threshold values and computed the overall speed in the PST as the mean over the speeds estimated with different threshold values.

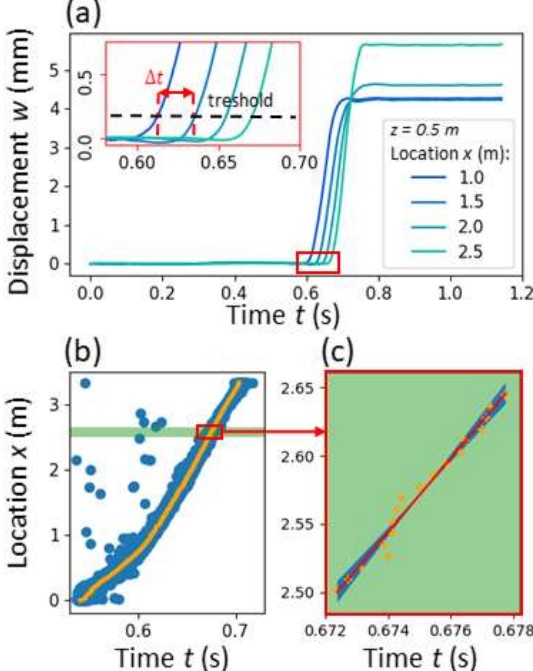

**Figure 4: (a) Slope normal displacement with time for four DIC-subsets along the PST. Colors indicate the $x$-location of the DIC-subsets. The inset shows the displacement threshold (dotted line) used to determine the time difference $\Delta t$ between the different DIC-**
**subsets. (b) Position of DIC-subset with time the threshold was passed. Blue dots show individual DIC-subsets while the orange dots show the median of all DIC-subsets with the same $x$-location. (c) is an inset of (b). $X$-location of DIC-subset with median time of threshold crossing (orange plus signs) for the green area (beam section) shown in (b). The red line shows the linear fit used to obtain crack speed. The 95% confidence interval, used for uncertainty estimation, is shaded in blue.**

Second, we used the normal strain from weak layer DIC-subsets. Similar to the displacement-based method, we first smoothed
the strain curves $\varepsilon_{zz,i}(t)$ (Savietzky-Golay filter, window size 31 DIC-subsets and order 3) before we applied a threshold of -0.01 to the strain in the weak layer. We removed outliers by neglecting time stamps below the 5[th] percentile and above the 95[th] percentile. For the remaining time stamps we calculated the median of all DIC-subsets with the same x-location before we estimated crack speed again as the slope of linear fits to overlapping moving windows (25 cm, step size 2.5 cm). The superscript "strain" indicates this methodology as in $c^{\text{strain}}$.

Third, we calculated crack speed by cross-correlating the slope normal acceleration $\ddot{w}(t)$ of the DIC-subsets. For a given beam section of width $\Delta x = 30$ cm, we cross-correlated the slope normal acceleration curves $\ddot{w}(t)$ of all DIC-subset pairs with same $z$-location (without repetition) to obtain time lags $\Delta t$ with pair spacing $\Delta d$ (Figure 5). Crack propagation speed was then determined by a linear fit for data pairs of time and pair spacing, $\Delta t$ vs. $\Delta d$. Thus, this approach allowed us to obtain a crack speed estimate $c^{\text{corr}}(x)$ for a specific beam section similar the way as described for the first and second method (Figure 4,
bottom graphs), but without the necessity of choosing a threshold. As an uncertainty estimation, again the 95 % confidence

interval of the fit was chosen. For all three methods, crack speed over the entire PST experiment was then simply taken as the mean of $c(x)$.

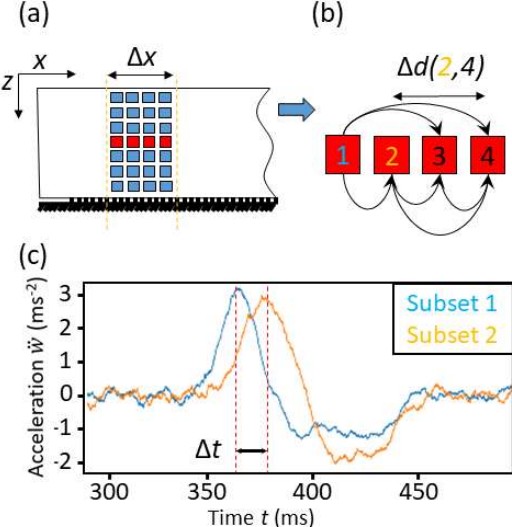

**Figure 5: (a) For the correlation based speed estimation we used beam sections, which contain vertically and horizontally aligned DIC-subsets (squares). (b) For each line (same $z$-location in beam section) of DIC-subsets, all possible combinations of two DIC-subsets formed a couple (black arrows) with a distinct distance $\Delta d$. (c) The normal accelerations $\ddot{w}(t)$ of the couples were used for temporal cross-correlation to derive the time lag $\Delta t$ between the DIC-subsets.**

**Touchdown distance**

As the crack propagates through the PST beam, the slab bends before coming into contact with the crushed weak layer. Considering a DIC-subset in the slab, as long as it is ahead of the crack tip, it has a slope normal velocity $\dot{w}(t)$ of zero. The velocity then increases as the crack passes underneath. Finally, as the crack has passed, the slope normal velocity returns to zero. We therefore defined the length $\lambda$ as the distance between DIC-subsets at rest before and after the collapse. To estimate $\lambda$ we averaged normal velocities $\dot{w}(t)$ of all vertically aligned DIC-subsets in the slab for each time step (slab columns, solid lines in Figure 6). Then, we performed spatial smoothing along $x$ (Savitzky-Golay, window 61, order 3, red dashed lines in Figure 6) before we applied a threshold to the vertical velocity $\dot{w}(t)$ ($1.9 \times 10^{-4}$ ms⁻¹, standard deviation before crack propagation). The touchdown distance $\lambda$ is then the distance over which columns exceed the velocity threshold (Figure 6). Uncertainty was estimated by taking the difference in $\lambda$ if we would have taken a three times larger threshold.

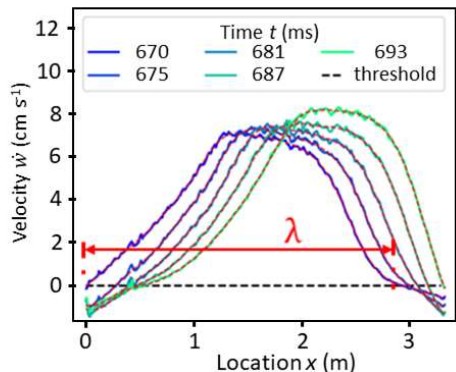

**Figure 6: Raw (solid lines) and smoothed (red dashed lines) vertical velocity $\dot{w}(t)$ of slab columns with x-location for different times while the crack propagates through the PST For each time step, the touchdown distance ($\lambda$) was estimated as the distance between the two closest slab columns that are still at rest or have just come to rest. One representation of the touchdown-distance at time 670 ms is indicated in red.**





## 3. RESULTS

We analyzed three Propagation Saw Test (PST) experiments performed on three different days in January 2019 at the same site and on the same weak layer consisting of surface hoar. We did not note any changes of the weak layer during these 10 days in terms of thickness and grain size (Table 1). The thickness of the slab, on the other hand, increased from 23 to 53 to

83 cm. At the same time the load due to the slab increased from 318 to 708 up to 1217 Pa while the mean density of the slab did not change much (between 136 and 149 kgm$^{-3}$). To investigate variations in snow properties along the PST beam, we performed four to five SMP measurements along the beams (Figure 7). The mean penetration resistance of the slab increased from 90.6 to 116 to 220 mN. Generally, the heterogeneity within the three PSTs was negligibly small and the SMP measurements were in good agreement with manual profiles (e.g. PST3 in Figure 7).

The critical cut length increased continuously from PST1 to PST3 and crack propagation was very different. Indeed, PST1 resulted in crack arrest due to a slab fracture (SF), PST2 showed crack arrest (ARR) without slab fracture and in PST3 the crack propagated to the very end of the beam: full propagation (END).

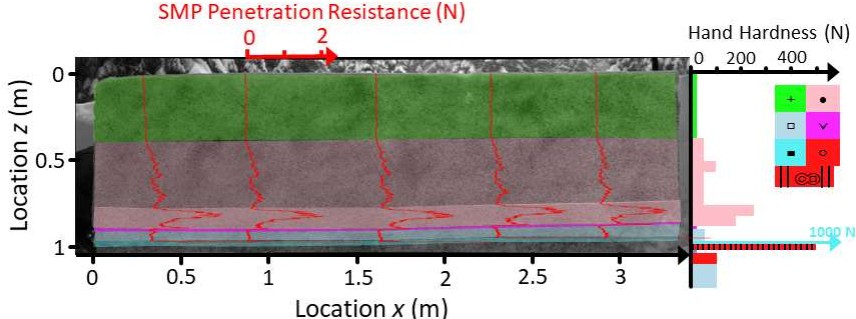

**Figure 7: Penetration resistance measured with the SMP (red lines plotted on the PST image) at the location where they were measured in PST3. The coloring of the background corresponds to the layers identified in the manual profile (right side). The manual**
**profile shows hand hardness index and grain shape indicated by colors (Fierz et al., 2009).**

**Table 1: Overview of the three propagation saw tests containing field notes about the PST, slab and weak layer characteristics from the manual and SMP profiles as well as video parameters and DIC analysis parameters.**

| PST | PST number | 1 | 2 | 3 |
|---|---|---|---|---|
| | Date | 4 Jan 2019 | 10 Jan 2019 | 13 Jan 2019 |
| | result | slab fracture | crack arrest | full propagation |
| | beam length (m) | 2.3 | 3.3 | 3.3 |
| | critical cut length (m) | 0.205 | 0.22 | 0.33 |
| slab | slab thickness (m) | 0.23 | 0.565 | 0.74 |
| | mean density (kgm$^{-3}$) | 138 | 136 | 149 |
| | mean penetration force (mN) | 90.6 | 116.0 | 220.1 |
| weak | grain type | ∨ (SH) | ∨ (SH) | ∨ (SH) |
| layer | grain size max (mm) | 15 | 15 | 15 |
| | grain size avg (mm) | 10 | 10 | 10 |
| video | frame rate (frames s$^{-1}$) | 20,000 | 14,000 | 7,000 |
| | image height (pixel) | 256 | 400 | 400 |
| | lens aperture value | 2.8 | 2.8 | 2.8 |
| | lens focal length (mm) | 24 | 24 | 24 |
| DIC | subset size ( pixel) | 9 | 12 | 12 |
| | subset step size ( pixel) | 3 | 3 | 3 |

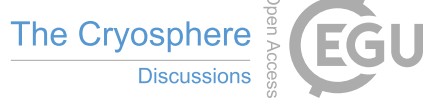

### 3.1. Displacement and strain

While the visual observation of the PSTs allowed to detect the outcome of PST1 as SF and PST3 as END, we could not discern

the result of PST2. We noticed crack propagation did not reach the far end, without seeing an obvious place where the crack had stopped. Only after consulting the displacement and strain fields it became clear that PST2 resulted in ARR.

For PST3, that resulted in full propagation (END), z-displacement increased with time, starting at the sawing end. Total z-displacement after weak layer fracture was lowest between positions 0.8 m < x < 2 m along the beam. For x > 2 m the total z-displacement increased up to 10 mm – about twice the z-displacement measured for smaller x. This large z-displacement is

attributed to a secondary crack propagating in the opposite direction. This is particularly clear when looking at the strain field after crack propagation (Figure 8 c3 and Appendix A for the temporal evolution). This secondary crack propagated when the initial crack in the surface hoar weak layer at z = 0.8 m reached the far end of the beam. The propagation of this secondary crack is clearly visible in the x-displacement (Figure 8 c2). For x < 2 m, x-displacements returned to zero after crack propagation. For x > 2 m, on the other hand, a residual positive x-displacement remained after crack propagation (e.g.,

t = 0.8 s). This residual x-displacement is grouped into three sections that align very well with the two slab fractures stopping this reverse propagating crack (Appendix A). Thus, although in the field the crack propagation in PST3 was recorded as a simple END, the displacement and strain data clearly show that the crack propagation dynamics were more intricate, and a combination of END, SF and ARR.

In PST1 the slab fracture (SF) was visible in the field and is clearly reflected in the measurements as well, in particular in the

strain field (Figure 9). When the saw reached $r_{saw}$ = 20 cm a crack within the weak layer started propagating (t = 90 ms in Figure 9 b). A tensile crack in the slab then opened at $x_{SF}$ = 32 cm (t = 120 ms in Figure 9 c) and stopped crack propagation in

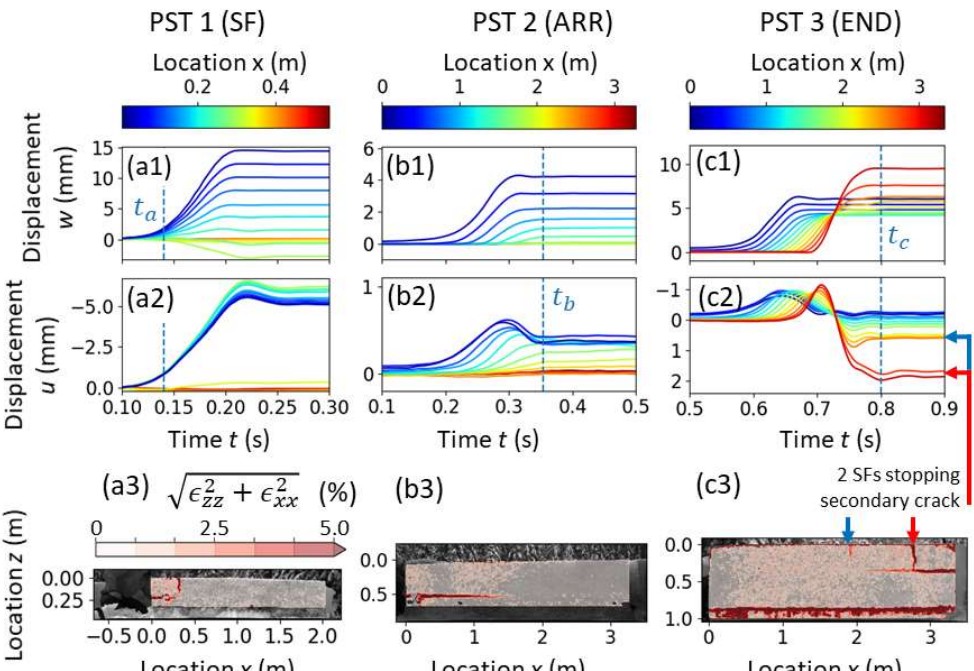

**Figure 8: Displacement and strain fields for the three different PSTs. (a) Slab fracture, (b) Arrest (c) Full propagation. The deformation of the slab with time is shown as displacement curves w (vertical, along z, first row) and u (horizontal, along x, second row) measured at various locations x along the PST beam. The third row shows the strain fields at the respective times $t_a$, $t_b$ and $t_c$, indicated with the dashed lines in the upper two rows. Strain concentration around cracks is clearly visibly.**


the weak layer ($t$ = 160 ms in Figure 9 d). As the slab fractured, $z$-displacement left of the SF ($x < x_{SF}$) exhibited downward displacement (positive $z$) whereas columns very close to $x_{SF}$ show upward displacement (negative $z$-displacement, Figure 8 a1). This suggests that the portion of the slab that became detached rotated with a rotation point close to $x_{SF}$. The $x$-displacement (mean along $z$) of all DIC-subsets in the detached part of the slab was very similar (blue and green lines in Figure 8 a2).

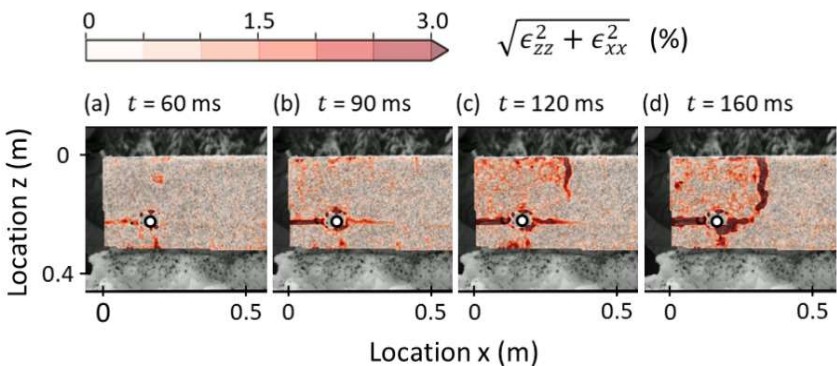

**Figure 9:** Magnitude of normal strains (colors) in the first half meter of PST1 at four different time steps: (a) no weak layer cracking ahead of the snow saw (black circles with white dot), (b) crack propagation ahead of the saw, (c) appearance of a crack in the slab propagating downwards from the snow surface, and (d) both cracks have connected.

PST2 resulted in crack arrest (ARR) and the strain at time $t = t_b$ at the end of crack propagation was largest in the weak layer up to a distance of around $x$ = 1.6 m (Figure 8 b3). That the crack propagated to this point was also clearly visible in the vertical displacement of the DIC-subsets in the slab. Indeed, the end displacement ($t > t_b$) of the slab decreased continuously with increasing $x$ until no vertical displacement was observed for $x > 1.6$ m (Figure 8 b1). The horizontal displacement $u(t)$, on the other hand, extended beyond the arrested crack tip (1.6 m < $x$ < 2 m), as expected for a bending slab. Interestingly, for $x < 1.2$ m, $u(t)$ decreased after reaching a maximum value before $t_b$, suggesting that the slab experienced support from the disaggregated weak layer and substrate. This recovered support introduces a bending moment, which acts opposite to the bending moment of the solely free hanging beam end.

### 3.2. Mechanical properties

The effective elastic modulus $E_{sl}^{VH}$ of the slab was obtained by fitting the solution of Heierli et al. (2008a; eqs 1 & 5) (Figure 3a) to all cut lengths $r_{saw} < r_c$. It predicts $E_{sl}^{VH}$ to be 14.5 MPa (Table 2 and Figure 10a, orange triangle with largest $r_{saw}$). Applying the RW method at $r_{saw} = r_c$ the elastic moduli $E_{sl}^{RW}$ and $E_{wl}^{RW}$ were 5.4 MPa and 0.12 MPa respectively. Hence, the RW method predicts a lower elastic modulus of the slab as the VH method, but is still higher than the SMP based modulus $E_{sl}^{BR}$ = 2.7 MPa (mean over 5 measures along the beam, std = 0.21 MPa).





**Table 2: Effective elastic modulus of the slab and weak layer ($E_{sl}$ and $E_{wl}$) as well as the weak layer specific fracture energy $w_f$ derived from the VH, RW, and BR method for PST3.**

|      | $E_{sl}$ (MPa) | $E_{wl}$ (MPa) | $w_f$ (Jm$^{-2}$) |
|------|------|------|------|
| VH   | 14.5 | -    | 0.28 |
| RW   | 5.4  | 0.12 | 0.31 |
| BR   | 2.7  | -    | 0.31 |

To verify the robustness of the derived elastic modulus $E_{sl}^{VH}$, we progressively increased the upper boundary of the fit interval $r_{max}$ from $r_{max} = 20$ cm to $r_c$. The elastic modulus initially decreased rapidly from 49 MPa to 20 MPa for $r_{max} = 25$ cm. Afterwards, the decrease was much slower with finally reaching a value of 14.5 MPa at $r_{max} = r_c$ cm. Also the RW method

predicts higher elastic moduli for short cut lengths (Figure 10 a, blue triangles). Compared to the VH method (orange) the RW method estimates elastic moduli of the slab and weak layer very consistently for cut lengths 15 cm $< r < r_c$ (Figure 10 a).

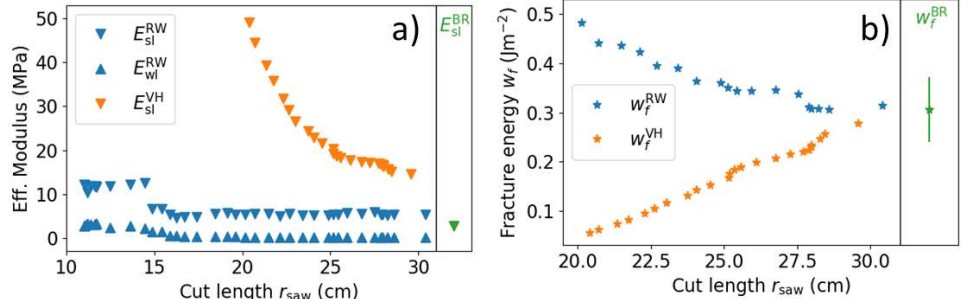

Figure 10: (a) Effective elastic moduli estimated with VH ($E_{sl}^{VH}$) and RW method ($E_{sl}^{RW}$, $E_{wl}^{RW}$) from different cut lengths. (b) Specific fracture energy estimated with RW method ($w_f^{RW}$, blue stars in b) and VH method ($w_f^{VH}$, orange stars) show contrasting behavior with increasing cut length. On the right side, for comparison, $E_{sl}^{BR}$ and $w_f^{BR}$ estimated from SMP signals are given in green. Whiskers

show the range of standard deviation.

All three methods estimated similar values of the specific fracture energy $w_f$. While the BR and the RW method predicted $w_f^{RW} = w_f^{BR} = 0.31$ Jm$^{-2}$ the VH method estimated a little lower value with $w_f^{VH} = 0.28$ Jm$^{-2}$ (Table 2). For the VH and RW method we quantified the robustness again by checking the trends with increasing cut lengths $r_{saw}$ (Figure 10b). Both methods showed contrasting trends with $r_{saw}$. Of course, to derive $w_f$ both models are evalutated at the critical cut length $r_{saw} = r_c$, but

the computation of $w_f$ needs $E_{SL}$ as well, and $E_{SL}$ is sensitive to the cut length it is based on.

### 3.3. Crack speed and touchdown distance

We used three different methods to estimate the crack speed and its evolution within the PST beam: the displacement, the strain and the cross-correlation approach. With the displacement and the strain approach the crack tip is located using threshold values, while no threshold is required for cross-correlating the acceleration curves. This similarity in methods is also reflected

in the similar crack speeds $c^{disp}$ and $c^{strain}$ determined in PST2 and PST3 (blue and green lines in Figure 11). Only in the first 20 cm of crack propagation ahead of $r_c$ were the values of $c^{strain}$ significantly lower than $c^{disp}$. Results obtained with the correlation method were very different, as values of $c^{corr}$ were significantly higher with also completely different trends throughout the PST experiments.

In PST2 the crack did not reach the far end of the beam, and hence values of crack speed could only be determined up to the

point where the crack arrested ($x = 1.6$ m, Figure 11 b). The speeds $c^{disp}$ and $c^{strain}$ exhibited the same overall behavior, and there was no clear trend in crack speed values through the PST experiment. Mean values were $c^{disp} = 9.1 \pm 0.9$ ms$^{-1}$ and


$c^{\text{strain}} = 6.9 \pm 2.6$ ms$^{-1}$. In contrast, mean $c^{\text{corr}}$ was $34.5 \pm 2.2$ ms$^{-1}$. It had an initial crack speed of $c^{\text{corr}}(x = r_c) = 46 \pm 3$ ms$^{-1}$, increased further and peaked at around $x = 0.4$ m at $56.4 \pm 1.9$ ms$^{-1}$. Afterwards it decreased steadily until $c^{\text{corr}}(x = 1.25$ m$) = 17.5 \pm 0.6$ ms$^{-1}$ before rising sharply again towards the crack arrest point.

**Table 3: Mean crack speeds of PST2 and PST3 estimated with the displacement ($c^{\text{disp}}$), strain ($c^{\text{strain}}$) and correlations ($c^{\text{corr}}$) based method.**

|  | $c^{\text{disp}}$ (ms$^{-1}$) | $c^{\text{strain}}$ (ms$^{-1}$) | $c^{\text{corr}}$ (ms$^{-1}$) |
|---|---|---|---|
| PST2 | $9.1 \pm 0.9$ | $6.9 \pm 2.6$ | $34.5 \pm 2.2$ |
| PST3 | $22.2 \pm 2.3$ | $17 \pm 5$ | $73 \pm 11$ |
| PST3(1m<$x$<2m) | $24 \pm 3$ | $21 \pm 5$ | $30.3 \pm 1.3$ |

For PST3 the mean speeds were $<c^{\text{disp}}>= 22.2 \pm 2.3$ ms$^{-1}$, $<c^{\text{strain}}>= 17 \pm 5$ ms$^{-1}$, and $<c^{\text{corr}}> = 73 \pm 11$ ms$^{-1}$. As in PST2, $c^{\text{disp}}$ and $c^{\text{strain}}$ exhibited a similar trend across the PST experiment. Initially, crack speeds slightly increased up to about $x = 1$ m, and remained rather constant thereafter (Figure 11 a, blue and green lines). The values of $c^{\text{corr}}$, on the other hand, were much

higher, especially at the beginning and the end of the experiment (Figure 11 a, orange line).

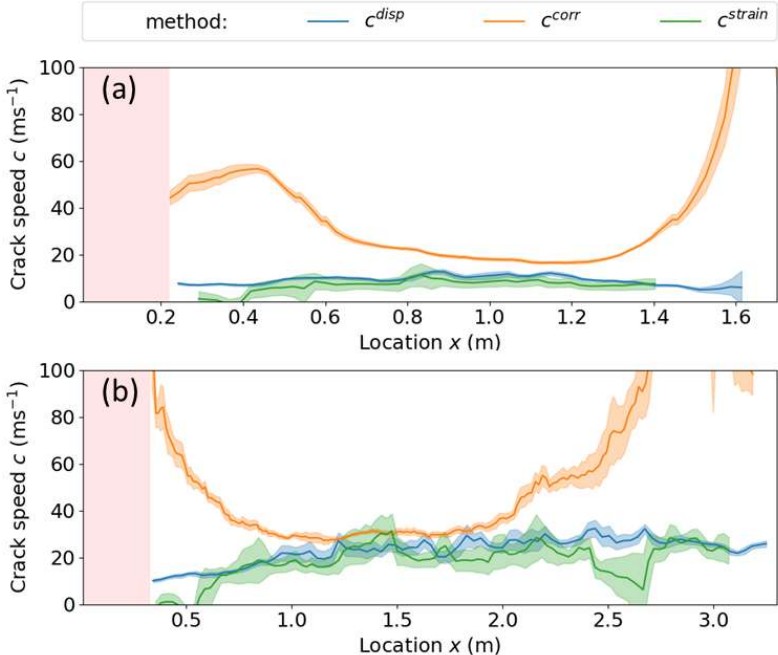

**Figure 11: Speed estimates along the beam for (a) PST2 and (b) PST3. The speeds are either based on the displacement (blue curve), the strain (green line) or the correlation method (orange line). The area of the beam that was saw cut ($x < r_c$) is shaded red. The 95% confidence interval was used to indicate the uncertainty and is depicted as a transparent region behind the corresponding lines.**

For PST3 it was possible to estimate the length of the touchdown distance $\lambda$ and its temporal evolution along the PST beam.

As DIC-subsets at the beginning of the PST beam came to a rest ($\dot{w}(t) < 1.9 \times 10^{-4}$ ms$^{-1}$) the velocity of DIC-subsets at $x = 2.9$ m started exceeding the threshold value, suggesting an initial touchdown distance of 2.9 m. As the crack propagated across the beam, $\lambda$ decreased to little less than 2.68 m before it slightly increased again towards the far end of the PST beam (Figure 12). As the touchdown distance appeared to be little shorter than the beam length, tracking was only possible within the last 40 cm of crack propagation.





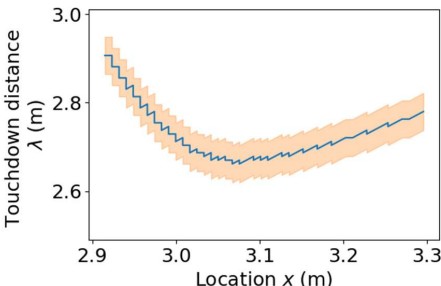

**Figure 12: Touchdown distance $\lambda$ of the propagating crack in PST3 with crack tip location (blue line). The orange transparent region indicates the uncertainty.**

## 4. DISCUSSION

We presented an experimental method to analyze the self-sustained crack propagation, i.e. the beginning of dynamic crack propagation in weak snowpack layers. In this phase we observed weak layer collapse and the associated slab bending, in accordance with many previous field studies (van Herwijnen and Jamieson, 2005;van Herwijnen et al., 2010;van Herwijnen and Birkeland, 2014;van Herwijnen et al., 2016b). However, compared to these previous studies that relied on particle tracking velocimetry (PTV), the digital image correlation method we used allowed us to observe these processes in much greater detail. For the first time, we were able to measure strain fields, showing strain concentrations in the area of the weak layer (Figure 8) as well as in the slab in experiments with slab fractures. The high frame rate of our high-speed camera combined with the much higher spatial resolution also allowed us to obtain detailed insight in changes in crack speed and touchdown distance with ongoing crack propagation.

When observing a PST experiment in the field, it is often very difficult to determine the exact location of crack arrest, distinguish between crack arrest far away from $r_c$ and full propagation, or determine whether a slab fracture occurred when the crack arrested. The unusual results obtained for PST3, where full crack propagation occurred, followed by a secondary fracture in a different weak layer, were not recognized in the field. With PTV, the method used in previous studies to investigate crack propagation in PSTs, the interpretation of the observed differences in the displacement curves would be ambiguous (Figure 8, c1 to c3). However, the strain field obtained with DIC clearly highlighted the presence of the secondary crack propagating in the opposite direction in a weak snow layer consisting of precipitation particles, which was closer to the snow surface. This secondary fracture was very likely triggered by the sudden drop of the slab after the crack had propagated through the tested weak layer. The secondary crack was stopped by two slab fractures (Appendix A). These results clearly highlight the advantages of DIC to investigate intricate subtleties occurring in PST experiments and resolve the processes during crack propagation in great detail.

Despite the increased detail obtained with DIC, it was not possible to measure absolute values of strain in the weak layer. The DIC-subset size ($\approx 3$ cm) was still larger than the vertical extent of the weak layer ($\approx 1.5$ cm). Values of strain should thus be considered as an average strain over an area with high strain occurring in the weak layer and areas with rigid body motion (portion of the slab, visible in the DIC-subset) or even areas without motion (portion of the substrate. Reducing the field of view of the camera increases spatial measurement resolution, thus by taking close-ups of the weak layer it is theoretically possible to reduce the DIC-subset size to less than the extent of the weak layer. However, as snow is a porous material, it consists of interconnected ice crystals. The thickness of surface hoar layers is often on the size scale of individual crystals. Therefore, the concept of a continuum strain in the weak layer does not exist at this scale, since strain distribution is locally very heterogeneous within the ice matrix. In addition, an appropriate speckling of the measuring surface then becomes difficult, as single crystals would have to be speckled. The strain measurements obtained with DIC therefore show strain localization





indicative of crack formation and propagation, but cannot accurately measure the exact deformation behavior within the weak layer.

### 4.1. Crack speed and touchdown distance

The high frame rate of our recordings allowed time derivatives of the displacements, and thus to obtain speed and acceleration of the DIC-subsets. In the past, crack speed and touchdown distance were solely based on displacement data (Bair et al., 2014; van Herwijnen et al., 2010; van Herwijnen and Jamieson, 2005). Exploiting speed and acceleration data of the slab allowed us to estimate crack speed and touchdown distance differently. To estimate these quantities, the position of the crack tip must be known at all times. For opening cracks, the position of the crack tip is obviously the place where the material separates. For closing cracks, as is the case in weak layer failure in snow, no generally valid definition exists. In order to estimate its influence, different definitions have been applied to estimate crack propagation speed.

We attribute differences between crack speeds found in the experiments to the dynamics of crack propagation. With dynamics, we understand that with crack extension the geometric configuration of the PST, the load type and strain rate, as well as the boundary conditions change. In turn, these are factors that influence the crack growth itself. It is therefore to be expected that the displacements of the slab, and thus all the variables derived from it, such as DIC-subset speed and acceleration, will change during crack propagation. For instance, in Figure 8a and 9b the displacement curves change their shape with $x$. The different methods to estimate speed are influenced by different aspects of this change of shape. For example, $c^{\mathrm{disp}}$ is sensitive to how rapidly the displacement curve increases at the beginning up to the threshold. The speed $c^{\mathrm{corr}}$, however, is influenced by the whole displacement curve by taking into account the change in shape using the entire curvature of the displacement curve. These shape changes are most pronounced near both ends of the PST. Of course, there, the boundary conditions also change and edge effects are to be expected (Bair et al., 2014). Hence, different methods may well yield different crack speeds, especially as long as the dynamics change, and no steady state propagation is reached. Looking more closely at the drivers of the dynamics, two effects can be separated. At the saw end of the beam, from where the crack starts propagating, crack extension causes an increasingly longer free-hanging slab as long as the new crack faces are not in contact yet. This leads to changes in magnitude and loading angle at the location of the crack tip. As the crack approaches the far end of the beam, changes in magnitude and loading angle at the crack tip develop again as the bending moment is forced to zero at the end of the beam. These changes naturally also change the shape of the displacement curves and not least, the crack speeds. In the middle part of the beam, edge effects are less pronounced. Here, possible drivers for crack propagation dynamics are strain rate effects and smaller geometric changes, e.g. changes in touchdown distance. Nevertheless, as long as crack propagation is not in a steady state, displacement curves $w(t)$, measured in the slab, change along the PST beam (increasing $x$-location). These changes can potentially explain the offset between the measurement methods. Therefore, very long PST experiments would be needed to clarify the existence of steady-state crack propagation (Heierli, 2005). In such experiments, the two prominent effects (close to beam ends and far from beam ends) of dynamics should be more clearly separated. This should allow for measurements of constant crack speed far from the beam edges, no matter which method is applied. Summing up, we found that the correlation based method ($c^{\mathrm{corr}}$) was very sensitive to changes in propagation dynamics and may be more suitable for investigating edge effects than for determining reliable measures of crack speed. Crack speed estimates of $c^{\mathrm{strain}}$ and especially $c^{\mathrm{disp}}$ were more robust and therefore more suitable to measure crack propagation speed in a PST.

While we were not able to observe steady-state crack propagation within our PSTs, the measured speeds can be compared with theoretical predictions based on a solitary wave model. Heierli (2005) formulated simple expressions for the crack propagation speed and wavelength of a steady-state collapse wave. Applying this model, for PST3 we obtained a wavelength of 2.7 m that travels with 35 ms⁻¹. Considering the speeds around the middle of the beam (1 m < x < 2 m), the mean speeds we observed were $c^{\mathrm{corr}} = 30.3 \pm 1.3$ ms⁻¹, $c^{\mathrm{strain}} = 21 \pm 5$ ms⁻¹ and $c^{\mathrm{disp}} = 24 \pm 3$ ms⁻¹. Thus, with the three methods we estimated lower speeds than the model for steady state crack propagation predicts. Whereas the predicted wavelength and touchdown distance we





observed were in good agreement. Again, considering the touchdown distance in the middle of the PST beam, we found a touchdown distance of $2.68 \pm 0.04$ m. So far, a single measurement of touchdown distance was reported by Bair et al. (2014). They measured much longer distances than theoretically predicted and attributed the discrepancy mostly to the model assumption, namely that the slab is in free fall motion while collapsing. While their measurement did not allow quantifying this assumption, we were able to derive slab accelerations. We found maximum downward accelerations of around $\max[\ddot{w}(t)] = 3$ ms$^{-2}$, that clearly demonstrate that the slab is not experiencing free fall (Appendix A, video A1).

### 4.2. Elastic modulus and weak layer specific energy

In our study all estimates of effective modulus of the slab were of the same order of magnitude; the ratios were approximately $E_{\mathrm{sl}}^{\mathrm{BR}} \approx \frac{1}{2} E_{\mathrm{sl}}^{\mathrm{RW}} \approx \frac{1}{5} E_{\mathrm{sl}}^{\mathrm{VH}}$. Comparing the moduli derived from the displacement fields, we consider the ones obtained with the RW method as probably the most appropriate. They do not rely on Clapeyron's theorem, seem to be stable with increasing cut lengths and the visual similarity between the experimentally determined data and the applied model seems to be good (Figure 3). Moreover, the RW method also allows estimating the elastic modulus of the weak layer in a PST experiment under the assumptions of isotropy.

For the weak layer specific fracture energy a couple of measurement methods already existed (Schweizer et al., 2011). The results obtained with the RW method were in good agreement with the estimates of the other methods $(w_f^{\mathrm{RW}} = w_f^{\mathrm{BR}} = 1.1\ w_f^{\mathrm{VH}})$.

### 5. CONCLUSIONS

We recorded PST experiments using a portable high-speed camera. By applying a speckling pattern on the entire sidewall of a PST column, we then used digital image correlation (DIC) to derive the displacement and strain of the slab, and the strain across the weak layer.

From displacement and strain fields we derived two independent estimates of crack speed. In addition, we computed crack speeds by correlating the downward acceleration of the slab in time. Our results suggest that crack speed can be reliably derived with both threshold based approaches as the determined values were close. Values obtained with the correlation based technique were however susceptible to shape changes of the displacement curves. Hence, values from the correlation-based technique resulted in larger variations in speed along the PST beam. In general, with our measurement setup and analyses changes in crack propagation speed at the scale of a PST beam can be derived.

From the downward velocity field of the slab we estimated the touchdown distance and its change while the crack propagated through the PST. The one result we have, is in good agreement with theoretical predictions from a solitary wave model. The model assumption, however, that the slab is in free fall behind the crack tip, needs to be refuted based on the presented measurements of slab acceleration.

Crack speed and touchdown distance were both affected by edge effects on both PST ends, from the saw end as well as from the free end. These edge effects require much longer PST beams to study the propensity of sustained crack propagation.

While we measured the evolution of strain over the weak layer during crack propagation, it was not possible to determine the true strain within the weak layer due to limitations in the measurement setup. The spatial resolution is still too low since one strain window involves slab or substrate regions adjacent to the weak layer. In the future, we plan to increase the spatial resolution by filming close-ups around the weak layer, even though a measurement of true strain within the weak layer will probably not be feasible.

Furthermore, the increased spatial resolution of our DIC setup offered an alternative method for deriving the effective elastic modulus of the slab from the displacement field. Compared to already established methods, the method based on the model

https://doi.org/10.5194/tc-2020-360





presented by Rosendahl and Weissgraeber (2020) provided a robust estimation of the effective elastic modulus of the slab and,

in addition, of the weak layer.

Finally, we also computed the weak layer specific fracture energy using the different methods and obtained good agreement between them.

This study demonstrates the great potential of the experimental setup and DIC based analyses methods that in the future should allow for a deeper understanding of the dynamics of crack propagation at the slope scale, which ultimately determines

avalanche size.




**Appendix A:**

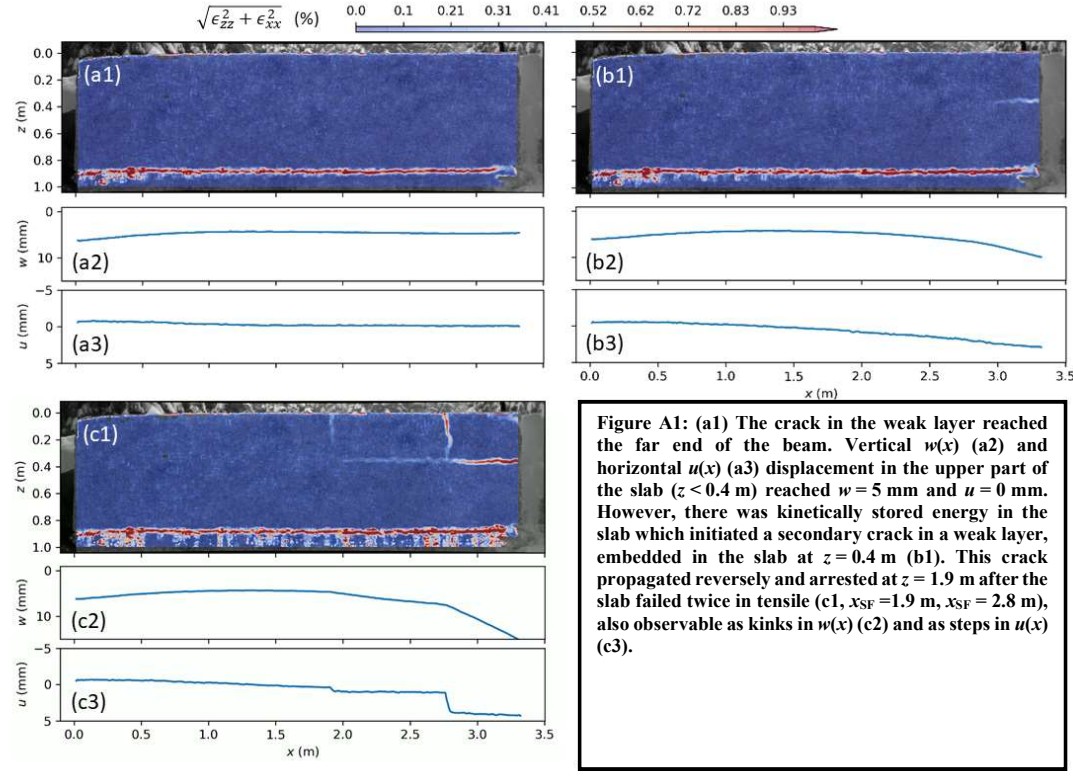

**Figure A1:** **(a1) The crack in the weak layer reached the far end of the beam. Vertical $w(x)$ (a2) and horizontal $u(x)$ (a3) displacement in the upper part of the slab ($z < 0.4$ m) reached $w = 5$ mm and $u = 0$ mm. However, there was kinetically stored energy in the slab which initiated a secondary crack in a weak layer, embedded in the slab at $z = 0.4$ m (b1). This crack propagated reversely and arrested at $z = 1.9$ m after the slab failed twice in tensile (c1, $x_{SF} = 1.9$ m, $x_{SF} = 2.8$ m), also observable as kinks in $w(x)$ (c2) and as steps in $u(x)$ (c3).**



A1:

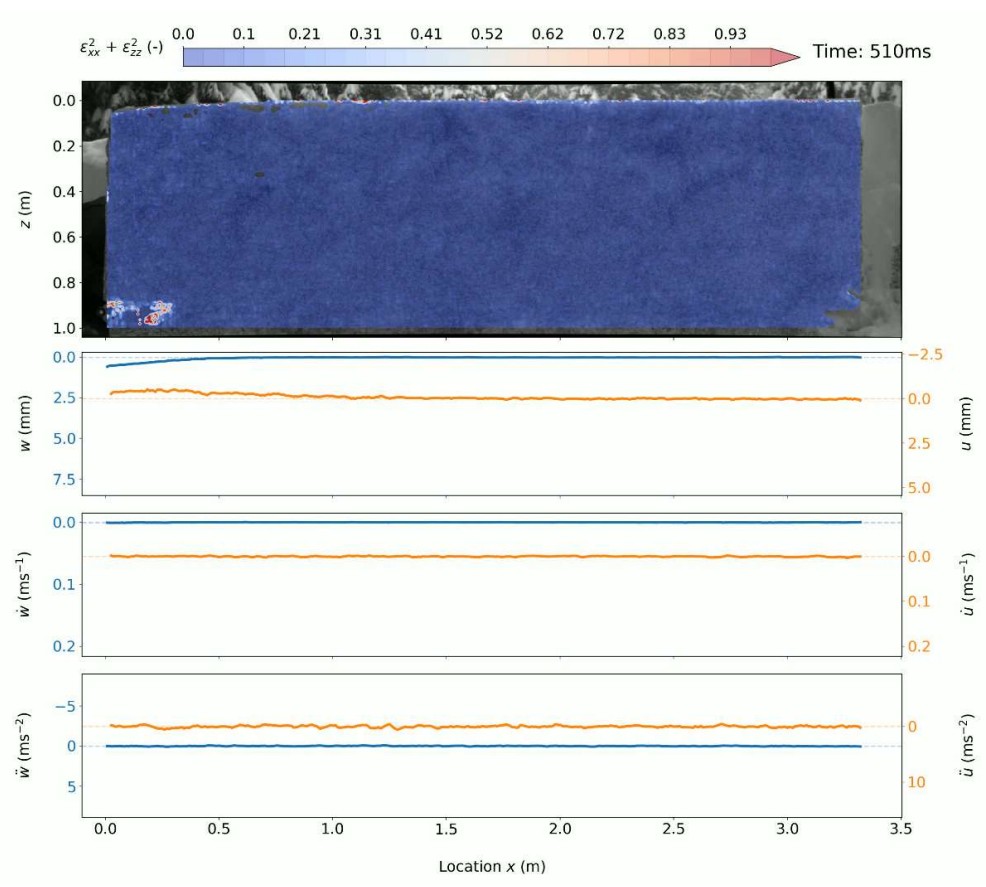

**Video A1: First frame of a video showing crack propagation in PST3. The video is available in the online supplementary material.**
**The strain magnitude shows a propagating strain concentration around the weak layer at $z = 0.9$ m (upper panel) while the vertical**
**displacement $w$ of the slab consecutively increases (blue line, second panel) and the horizontal displacement $u(z = 0.1$ m) indicates**
**slab bending (orange line, second panel). At the same time, the third and fourth panel show the (vertical, horizontal) velocity $(\dot{w}, \dot{u})$**
**and acceleration $(\ddot{w}, \ddot{u})$ computed as the first and second time derivative of the displacements, respectively. At a time $t = 740$ ms a**
**secondary crack in a weak layer, embedded in the slab at $z = 0.4$ m was initiated and propagated reversely after it is stopped as a**
**consequence of two slab fractures ($x_{SF} = 1.9$ m and $x_{SF} = 2.8$ m).**



## Appendix B:

We used the snow micro-penetrometer (SMP) to assess variations in snow properties along the PST beams. From the SMP signal an effective elastic modulus of the slab and the weak layer specific fracture energy can be derived. These two values were included in chapter 3.2. Additional, SMP based, parameters of PST3 are provided in Table A2.1 below.

**Table B.1: Values derived from SMP signals that were measured along PST3. Snow instability metrics include failure initiation criterion (INI), critical crack length (m) (PRO) and slab tensile criterion (TCR). Snow mechanical properties include weak layer shear strength (Pa) (TAU_p), specific weak layer fracture energy (Jm$^{-2}$) (WF_wl), effective slab modulus (MPa) (E_effslab), slab thickness (m) (HSLAB) and average slab density (kgm$^{-3}$) (RHOSLAB) as previously computed by Reuter and Schweizer (2018).**

| FILE | INI | PRO | TCR | TAU_p | WF_wl | E_effslab | HSLAB | RHOSLAB |
|---|---|---|---|---|---|---|---|---|
| 264 | 5.137 | 0.3 | 0.43 | 801.9 | 0.35 | 2.96E+06 | 0.81 | 155.85 |
| 265 | 4.159 | 0.18 | 0.55 | 602.1 | 0.2 | 2.37E+06 | 0.82 | 143.56 |
| 266 | 4.211 | 0.26 | 0.55 | 627.7 | 0.3 | 2.64E+06 | 0.84 | 146.22 |
| 267 | 3.961 | 0.27 | 0.56 | 621.4 | 0.31 | 2.70E+06 | 0.85 | 147.11 |
| 268 | 4.264 | 0.3 | 0.41 | 694.9 | 0.37 | 2.73E+06 | 0.84 | 147.57 |

*Data availability.* The high-speed recordings as well as the analysis routines are available upon request from the corresponding author.

*Supplement.* The supplement related to this article is available on-line.

*Author contributions.* JS and AH acquired the funding for the project leading to this publication. Together with BB and GB they designed the experimental setup before BB and AH carried out the experimental work. BB has developed and undertaken the processing of the high-speed recordings. BR performed the SMP analysis and JD supervised all parts of the project. The manuscript was written by BB with input from all authors.

*Competing Interests:* The authors declare that they have no conflict of interest.

*Acknowledgements:* Achille Capelli, Christine Seupel, Collin Lüond, Alexander Hebbe and Simon Caminada assisted with fieldwork.

*Financial support:* The Project is funded through the Swiss National Science Foundation (grant 200021_169424).

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
