# Peer review of "Dynamic crack propagation in weak snowpack layers: Insights from high-resolution, high-speed photography"

_The Cryosphere, 2020_

## Referee Comment (RC1)

rosendahl@ismd.tu-darmstadt.de, mail@2phi.de

The manuscript presents a methodology for full-field measurements of snowpack displacements using digital image correlation. The work opens numerous possibilities for the extraction of snowpack properties. Among these, the authors discuss ways to obtain an effective homogenized elastic modulus of the slab, the weak layer fracture toughness and the speed of cracks running in the weak layer. The study focuses on the comparison of different methodologies using three representative examples.

The paper makes a significant contribution towards the understanding and characterization of fracture mechanical processes that lead to slab avalanche release. However, I have a serious concern regarding the derivation of the weak-layer fracture energy from SMP signals, denoted $w_f^{\mathrm{BR}}$.

The manuscript cites Reuter and Schweizer (2018) [doi: 10.1029/2018GL078069] for a

description of the approach. In this work, however, I find no explanation of the methodology. Instead, I am referred to Reuter et al. (2018) [doi: 10.16904/envidat.40], which, again, does not clarify the procedure. In the accompanying README file I am referred to publication: Reuter et al. (2015) [doi: 10.5194/tc-9-837-2015], Eq. (4). Here, the fracture energy is obtained from the integration of the SMP force signal over certain windows and subsequent selection of the minimum value within the weak layer:

$$w_f = \min_{\text{WL}} \int_{-\frac{w}{2}}^{+\frac{w}{2}} F \mathrm{d}z, \tag{1}$$

where $w$ is the windows size and $F$ the penetration resistance. From the publication I understand that $w$ is of dimension length and $F$ of dimension force (e.g., Reuter et al. (2015) [doi: 10.5194/tc-9-837-2015], Figure 3). This yields units of Nm (energy) for $w_f$ when it should be N/m = J/m$^2$ (energy per unit area).

The following thought experiment raises another concern about the above equation (1). Imagine we probe the same weak layer (with the same fracture energy) with an SMP of twice or half the original diameter. The former should yield a larger resistance $F$, the latter a smaller one. Evaluating all three signals (original, double, and half diameter) with the same window size will yield three different fracture toughnesses of the same weak layer. Which one is correct?

Moreover, Reuter et al. (2015) [doi: 10.5194/tc-9-837-2015] refer to Reuter et al. (2013) [url: http://arc.lib.montana.edu/snow-science/objects/ISSW13_paper_O2-02.pdf] regarding the validation of the methodology surrounding the above equation (1). Here, the accuracy of the method is checked using an approach similar to the VH method used in the present work. However, in the present manuscript, the VH method is deemed unfit for the derivation of $w_f$, for instance because of its inability to model the measured strain energy (Figure 3a) or its inability to correctly account for the slab's Young modulus (Figure 10a). I encourage the authors to comment on this contradiction because I cannot understand the details of the procedure used by Reuter et al. (2013)

since no equations are given.

Concluding my concerns surrounding $w_f^{\mathrm{BR}}$, I specifically ask for clarification of the following:

1. Please explicitly explain (in the manuscript) how the fracture toughness $w_f^{\mathrm{BR}}$ is derived from SMP signals including corresponding equations and dimensions.

2. Please comment on the units of $w_f^{\mathrm{BR}}$ and Eq. (1) above.

3. Please comment on the issue different probe diameters regarding Eq. (1) above.

4. Please comment on whether I correctly understood the validation of Eq. (1) in Reuter et al. (2013) and the consequential contradiction.

These points should be clarified beyond doubt. If a comprehensive discussion of the methodology goes beyond the scope of the present work, I suggest omitting the SMP methodology for now. After all, its connection to high-resolution and high-speed photography is weak.

Aside from the above crucial points, I only have one other major remark:

5. Since you extract the external potential $V_{\mathrm{p}}$ directly from measured full-field data, Clapeyron's Theorem allows for direct identification of the total potential $V_{\mathrm{tot}} = V_{\mathrm{m}} + V_{\mathrm{p}} = V_{\mathrm{p}}/2$ and, hence, direct computation of the fracture energy $w_f = -\mathrm{d}V_{\mathrm{tot}}/\mathrm{d}r = -\mathrm{d}V_{\mathrm{p}}/(2\mathrm{d}r)$. No fitting to an analytical expression, only some form of signal processing of the experimental data shown in Figure 3a is required to compute the derivative.

Finally, I ask the authors to consider the following minor remarks:

6. The abstract devotes considerable attention to historical developments (lines 10–15) but does not include key findings of the manuscript. I suggest moving the historical perspective to the introduction and add key results such as determined crack speeds and fracture toughnesses – including the respective most suitable techniques for their identification.

7. (line 31) How does process of coalescence of subcritical failures work?

8. (line 47) Please motivate and discuss why and how crack speed is important.

9. (line 60) The touchdown length is not a material constant but depends, for instance, on the slab's bending stiffness and its density $\rho$. In order to give context to the listed absolute values, I suggest adding additional information.

10. (line 68) The statement is a bit misleading. The fracture energy itself is an independent fundamental material property and independent of other fundamental properties such as the elastic modulus. I assume what is meant is the following: because the method employs a certain model to compute $w_f$, and the model requires $E$ as and input, the back calculation will change if $E$ changes?

11. (line 130) Can you provide examples of used reference lengths?

12. (lines 157–162) The equation in line 162 only holds if $V_m$ and, hence, also $V_p$ in line 157 are defined per unit width. Please explicitly state (in an equation) how $V_p$ is determined. Is layering considered?

13. (line 175) Equation number missing.

14. (lines 413–414) Can you discuss possible reasons for this discrepancy? How does weak-layer rigidity or compliance affect crack speed?

15. (line 424) Clapeyron's Theorem is a fundamental law of mechanics and should not be brought in context with the limitations of certain models. Instead, I suggest

to explicitly repeat arguments for weaknesses of the VH method that were given around line 301.

Figures and Tables:

17. All images seem to have a low resolution and show compression artifacts. Is this a draft issue?

18. (Figure 1) Images are very small.

19. (Figure 2) Red text on gray picture is hard to read.

20. (Figure 10b) Why does $w_f^{\mathrm{RW}}$ decrease with $r_{\mathrm{saw}}$? I would expect the contrary. Is a constant Young's modulus chosen for each data point or does is change alongside $r_{\mathrm{saw}}$? I would suggest to use the "converged" effective modulus from Figure 10a (for both the VH and the RW methods) to calculate the fracture energies in 10b.

21. (Table 2) Again, please check the units of $w_f^{\mathrm{BR}}$.

22. (Table 4) The mean of $c^{\mathrm{corr}}$ suffers from (potential) inaccuracies towards the boundaries. Does it make sense to introduce a fourth column where the mean is evaluated on a more reasonable $x$-domain?

---

## Referee Comment (RC2)

[referee-annotated manuscript omitted]

---

## Author Comment (AC1)

Dear Dr. Rosendahl,

thank you very much for commenting and providing helpful suggestions on the manuscript. Below we have pasted your comments in blue, our point-by-point responses are given in black.

The manuscript presents a methodology for full-field measurements of snowpack displacements using digital image correlation. The work opens numerous possibilities for the extraction of snowpack properties. Among these, the authors discuss ways to obtain an effective homogenized elastic modulus of the slab, the weak layer fracture toughness and the speed of cracks running in the weak layer. The study focuses on the comparison of different methodologies using three representative examples. The paper makes a significant contribution towards the understanding and characterization of fracture mechanical processes that lead to slab avalanche release. However, I have a serious concern regarding the derivation of the weak-layer fracture energy from SMP signals, denoted wBRf .
The manuscript cites Reuter and Schweizer (2018) [doi: 10.1029/2018GL078069] for a description of the approach. In this work, however, I find no explanation of the methodology. Instead, I am referred to Reuter et al. (2018) [doi: 10.16904/envidat.40], which, again, does not clarify the procedure. In the accompanying README file I am referred to publication: Reuter et al. (2015) [doi: 10.5194/tc-9-837-2015], Eq. (4). Here, the fracture energy is obtained from the integration of the SMP force signal over certain windows and subsequent selection of the minimum value within the weak layer:

$$w_f = \min{}_{\mathrm{WL}} \int_{-\frac{w}{2}}^{+\frac{w}{2}} F \mathrm{d}z, \tag{1}$$

where w is the windows size and F the penetration resistance. From the publication I understand that w is of dimension length and F of dimension force (e.g., Reuter et al. (2015) [doi: 10.5194/tc-9-837-2015], Figure 3). This yields units of Nm (energy) for wf when it should be N/m = J/m2 (energy per unit area). The following thought experiment raises another concern about the above equation (1). Imagine we probe the same weak layer (with the same fracture energy) with an SMP of twice or half the original diameter. The former should yield a larger resistance F, the latter a smaller one. Evaluating all three signals (original, double, and half diameter) with the same window size will yield three different fracture toughnesses of the same weak layer. Which one is correct? Moreover, Reuter et al. (2015) [doi: 10.5194/tc-9-837-2015] refer to Reuter et al. (2013) [url: http://arc.lib.montana.edu/snow-science/objects/ISSW13_paper_O2-02.pdf] regarding the validation of the methodology surrounding the above equation (1). Here, the accuracy of the method is checked using an approach similar to the VH method used in the present work. However, in the present manuscript, the VH method is deemed unfit for the derivation of wf , for instance because of its inability to model the measured strain energy (Figure 3a) or its inability to correctly account for the slab's Young modulus (Figure 10a). I encourage the authors to comment on this contradiction because I cannot understand the details of the procedure used by Reuter et al. (2013) since no equations are given. Concluding my concerns surrounding wBR f , I specifically ask for clarification of the following:

1.  Please explicitly explain (in the manuscript) how the fracture toughness $w^{BR}_f$ is derived from SMP signals including corresponding equations and dimensions.

We regret that information was missing to understand the calculation of fracture energy derived from SMP measurements. We will include the necessary information in the Methods section of the revised manuscript:

*"As a third approach, we used SMP measurements. Effective elastic modulus $E_{sl}^{BR}$ was derived from SMP data as described by Reuter and Schweizer (2018), using the signal interpretation method suggested by Löwe and van Herwijnen (2012). Reuter et al. (2013) suggested a parametrization of the specific fracture energy based on the penetration resistance F(z). Using a moving window (size: w = 2.5 mm) to integrate F(z), they then defined the specific fracture energy as the minimum of the integral within the weak layer:*

$$w_f^{\text{SMP}} = A \ \min_{\text{WL}} \int_{-w/2}^{w/2} F \ dz \, ,$$

*where A is a fitting parameter. The integration has units energy (J) and relates to the work required to destroy the snow structure along the integration path. Specific fracture energy, however, has unit energy per area. Therefore, it is necessary to divide by an effective area, the fitting parameter A. While the effective area is unknown, it is likely larger than the cross section of the tip diameter (Johnson, 2003), and depends on snow structure(van Herwijnen, 2013) We therefore followed Reuter et al. (2019), and introduced a fitting parameter A to implicitly account for the unknown effective area. The fitting parameter was derived using a linear regression to PTV derived specific fracture energies (Figure 6 in Reuter et al. (2019)), resulting in A = 2.95 × 10³m⁻²). Which relates to a plausible effective cone area of 3.4 cm²(radius ≈ 1 cm)."*

2. Please comment on the units of w^BR_f and Eq. (1) above.

The unit discrepancy originated from the absence of the fitting factor *A* (see above, equation 2 in the reply to your comment 1). Its physical meaning and derivation will explicitly be stated (see reply above).

3. Please comment on the issue different probe diameters regarding Eq. (1) above.

With the fitting parameter A, accounting for the effective area, the equation to derive the fracture energy from SMP data now accounts for the probe size (see reply above).

4. Please comment on whether I correctly understood the validation of Eq. (1) in Reuter et al. (2013) and the consequential contradiction.

Given our reply above we hope that the issue is now clarified.
With the fitting parameter we introduced in Equation 2, it becomes clear now that the comparison to PTV-derived values in Reuter et al. (2013) and Reuter et al. (2019) served to parametrize the specific fracture energy on SMP signals. Reuter et al. (2013) and Reuter et al. (2019) showed a comparison of SMP- and PTV-derived values of the specific fracture energy. Rather than validating the accuracy of the SMP method, they discussed differences between different measurement methods. Currently lacking an alternative method for calibration, we used their PTV data to determine the factor *A* in equation 2.
Based on your comment 12 below, we applied corrections to the VH method (Fig 3a), which improved the fit of the mechanical energy. Nevertheless, the SMP-derived values are based on a parameterization derived from a linear regression with PTV-derived values. Once enough data are available, we will derive a parameterization using DIC data, or μCT data, or other future techniques, which may possibly provide more accurate $w_f$ data. We deem it valuable to provide the BR estimates and to compare them to more elaborate methods. Only then we know how the method performs and we can possibly calibrate the SMP approach to the best method in the future. For lack of alternative, the SMP currently remains the only

efficient experimental method to determine snow mechanical properties, such as the specific fracture energy, at many locations in the field.
These points will be discussed explicitly in the in the revised version of the manuscript.

These points should be clarified beyond doubt. If a comprehensive discussion of the methodology goes beyond the scope of the present work, I suggest omitting the SMP methodology for now. After all, its connection to high-resolution and high-speed photography is weak.

We agree that the main message of the paper is the potential of high-resolution DIC measurements. As the SMP is a relatively widely used measurement method, we decided to keep the SMP results. Moreover, we deem comparisons with exiting methods good practice. However, we substantially reduced the importance of the SMP derived values by only mentioning those in the text, and not in the figures anymore, and we explained the derivation of the fracture energy in greater detail, as mentioned above.

Aside from the above crucial points, I only have one other major remark:
5.  Since you extract the external potential Vp directly from measured full-field data, Clapeyron's Theorem allows for direct identification of the total potential Vtot = Vm + Vp = Vp/2 and, hence, direct computation of the fracture energy wf = dVtot/dr = dVp/(2dr). No fitting to an analytical expression, only some form of signal processing of the experimental data shown in Figure 3a is required to compute the derivative.

Thank you for your suggestion. As Heierli's formulation of the mechanical energy did not represent the measured data very well, we followed your suggestion by fitting an arbitrary function to our data. We only had two constrains for the function: 1) it should have a value of 0 for $r = 0$, and 2) the function should be monotonically decreasing with $r$. We used a simple power law function of the form $f(r) = -ar^b$ (FU) and refined the fitting window to 15 cm $< r_{max} < r_c$.
In the end, we assessed the quality of the VH and FU fit with the root mean squared error and found that the simple power law function FU represented the measured data better (RMSE$^{FU}$ = 0.007, RMSE$^{VH}$ = 0.013).
We will hence introduce the power law fit into the revised manuscript.

Finally, I ask the authors to consider the following minor remarks:

6.  The abstract devotes considerable attention to historical developments (lines 10–15) but does not include key findings of the manuscript. I suggest moving the historical perspective to the introduction and add key results such as determined crack speeds and fracture toughnesses – including the respective most suitable techniques for their identification.

We agree that the historical perspective was rather prominent and will revise the Abstract.

7.  (line 31) How does process of coalescence of subcritical failures work?

Coalescence of subcritical failures is part of our conceptual understanding of natural avalanche release (Schweizer et al., 2016). The formation of subcritical failures occurs at the microscale (scale of snow crystals and bonds, <<1 mm). At this scale two competing processes occur simultaneously (Capelli et al., 2018a): 1. Weak layer damage (meaning the breaking of bonds), and 2. Weak layer strengthening/sintering (meaning creation and strengthening of bonds). When the damage process dominates, more bonds break and a localized failure may develop, i.e. subcritical failures coalesce.

This damage process, aka failure events (bond breaking), manifests itself by acoustic emissions (Capelli et al., 2018b)

8. (line 47) Please motivate and discuss why and how crack speed is important.

We will motivate the importance of crack speed by pointing out that crack speed is an indicator of the crack propagation mode and may provide insight into an ongoing discussion about crack propagation in snow.

9. (line 60) The touchdown length is not a material constant but depends, for instance, on the slab's bending stiffness and its density $\rho$. In order to give context to the listed absolute values, I suggest adding additional information.

Indeed, the touchdown length is not a material property. We will provide the range of slab densities and slab thickness reported in Bair et al. (2014).

10. (line 68) The statement is a bit misleading. The fracture energy itself is an independent fundamental material property and independent of other fundamental properties such as the elastic modulus. I assume what is meant is the following: because the method employs a certain model to compute wf , and the model requires E as and input, the back calculation will change if E changes?

We agree with the reviewer that the statement was unclear. We will therefore reword the sentence to: *"This emphasizes a weakness of the method, the back-calculated specific fracture energy relies on the input of the elastic modulus. A parameter that contains large uncertainties, especially if it cannot be determined in-situ."*

11. (line 130) Can you provide examples of used reference lengths?

In all tests we acquired an image with a 2 m reference length fixed on the PST side wall. We will explicitly mention this.

12. (lines 157–162) The equation in line 162 only holds if Vm and, hence, also Vp in line 157 are defined per unit width. Please explicitly state (in an equation) how Vp is determined. Is layering considered?

We thank the reviewer for pointing out this mistake. Indeed, we did not define $V_p$ per unit width, resulting in wrong estimates of the elastic modulus and specific fracture energy of the VH method. We will correct this error and make the necessary changes throughout the revised manuscript.

13. (line 175) Equation number missing.

We will add the equation number.

14. (lines 413–414) Can you discuss possible reasons for this discrepancy? How does weak-layer rigidity or compliance affect crack speed?

In Heierli (2005), speed is proportional to the 4$^{th}$ root of the bending stiffness of the slab, which itself is directly proportional to the elastic modulus. Computing the speed with the VH and RW elastic modulus gives values of $c^{VH}$ = 25 ms$^{-1}$ and $c^{RW}$ = 35 ms$^{-1}$.

The model of Heierli assumes free fall motion of the slab during weak layer collapse, and therefore does not consider weak layer properties. Accounting for weak layer rigidity would therefore likely reduce the speed estimates.

15. (line 424) Clapeyron's Theorem is a fundamental law of mechanics and should not be brought in context with the limitations of certain models. Instead, I suggest to explicitly repeat arguments for weaknesses of the VH method that were given around line 301.

We will not mention Clapeyron´s Theorem anymore by changing the statement to:
*"Their (elastic modulus of RW method) estimation was stable with increasing cut length (**Fehler! Verweisquelle konnte nicht gefunden werden.**a) and the visual similarity between the experimentally determined data and the applied model seems to be good (**Fehler! Verweisquelle konnte nicht gefunden werden.**)."*

Figures and Tables:
16. All images seem to have a low resolution and show compression artifacts. Is this a draft issue?

We will improve resolution.

17. (Figure 1) Images are very small.

We will enlarge the three images with the limitation to fit everything in a single line. Since the intention of the figures is to present a schematic workflow of the processing, it was more important for us to keep a single line instead of showing the steps in detail.

18. (Figure 2) Red text on gray picture is hard to read.

We agree and will improve the figure.

19. (Figure 10b) Why does $w^{RW}_f$ decrease with rsaw? I would expect the contrary. Is a constant Young's modulus chosen for each data point or does is change alongside rsaw? I would suggest to use the "converged" effective modulus from Figure 10a (for both the VH and the RW methods) to calculate the fracture energies in 10b.

We do not see why the contrary should be expected. With a "perfect measurement" and a "perfect model" we would not expect any trend. Since this is, however, never the case, we investigated how strong the elastic modulus (moduli) varies when changing the fit interval ($r < r_{saw}$, VH method) or when taking another measured displacement field ($r = r_{saw}$, RW method). In a further step the weak layer fracture energies $w_f$ are derived from the elastic properties as:

$$w_f^{VH} = -\frac{d}{dr}V_m\bigg|_{r=r_c} \; ; \; w_f^{RW} = G_{I} + G_{II} = \frac{E_{wl}^{RW}}{2t}w_{RW}(r=r_c)^2 + \frac{E_{wl}^{RW}}{2t(v_{wl}-1)}u_{RW}(r=r_c)^2$$

Therefore, the variation of elastic properties propagates into $w_f^{VH}$ and $w_f^{RW}$. To illustrate that, Figure 10b shows how the variation of the elastic moduli affects the derived values of $w_f$.

Using a "converged modulus" would result in one specific fracture energy for each method. These energies are basically already given as the data points with largest $r_{saw}$ in Figure 10b.

To avoid misunderstandings, we will mention this explicitly:

"*Of course, to derive $w_f$ both models are evalutated at the critical cut length $r_{saw} = r_c$, but the computation of $w_f$ is based on $E_{sl}$ (and $E_{wl}$ for the RW method), and $E_{sl}$ is sensitive to changes of the fit interval ($r < r_{saw}$, VH method) or when taking another displacement field ($r = r_{saw}$, RW method).*"

20. (Table 2) Again, please check the units of wBR f .

This should now be clarified given our reply to your comment 2 above.

21. (Table 3) The mean of $c_{corr}$ suffers from (potential) inaccuracies towards the boundaries. Does it make sense to introduce a fourth column where the mean is evaluated on a more reasonable x-domain?

We are unsure whether you suggest to introduce a fourth column because you may oversee the last line in Table 3, in which the mean crack speed away from beam edges
(1 m < $x$ < 2 m) is shown.
Or, do you question if the last line in Table 3 is meaningful at all?
In that case we think it is useful to separate two regions and state mean crack speeds within. Our discussion also considers drivers of crack speed changes within these two regions and the regions are:
1.      Close to beam ends, where strong edge effects are to be expected.
2.      The middle part of the beam, where edge effects are less pronounced but still present as long as the crack is not in a steady state propagation.

**References**

Bair, E. H., Simenhois, R., van Herwijnen, A., and Birkeland, K.: The influence of edge effects on crack propagation in snow stability tests, The Cryosphere, 8, 1407-1418, 10.5194/tc-8-1407-2014, 2014.

Capelli, A., Reiweger, I., Lehmann, P., and Schweizer, J.: Fiber bundle model with time-dependent healing mechanisms to simulate progressive failure of snow, Physical Review E, 98, 023002, 10.1103/PhysRevE.98.023002, 2018a.

Capelli, A., Reiweger, I., and Schweizer, J.: Acoustic emissions signatures prior to snow failure, Journal of Glaciology, 64, 543-554, 10.1017/jog.2018.43, 2018b.

Heierli, J.: Solitary fracture waves in metastable snow stratifications, Journal of Geophysical Research, 110, F02008, doi: 02010.01029/02004JF000178, 2005.

Johnson, J. B.: A statistical micromechanical theory of cone penetration in granular materials, U.S. Army Corps of Engineers, Engineer Research and Development Center, Hanover NH, U.S.A., ERDC/CRREL Technical Report, ERDC/CRREL-TR-03-3, 32, 2003.

Löwe, H., and van Herwijnen, A.: A Poisson shot noise model for micro-penetration of snow, Cold Regions Science and Technology, 70, 62-70, 10.1016/j.coldregions.2011.09.001, 2012.

Reuter, B., and Schweizer, J.: Describing snow instability by failure initiation, crack propagation, and slab tensile support, Geophysical Research Letters, 45, 7019-7027, 10.1029/2018GL078069, 2018.

Reuter, B., Proksch, M., Löwe, H., van Herwijnen, A., and Schweizer, J.: Comparing measurements of snow mechanical properties relevant for slab avalanche release, Journal of Glaciology, 65, 55-67, 10.1017/jog.2018.93, 2019.

Schweizer, J., Reuter, B., van Herwijnen, A., and Gaume, J.: Avalanche release 101, Proceedings ISSW 2016. International Snow Science Workshop, Breckenridge CO, U.S.A., 3-7 October 2016, 1-11, 2016.

van Herwijnen, A.: Experimental analysis of snow micropenetrometer (SMP) cone penetration in homogeneous snow layers, Canadian Geotechnical Journal, 50, 1044-1054, 10.1139/cgj-2012-0336, 2013.

---

## Author Comment (AC2)

Dear Dr. Bair,

thank you very much for commenting and providing helpful suggestions on the manuscript. Below we have pasted your comments in blue, our point-by-point responses are given in black.

Line 10: "is" b.c. information is singular.

Thanks, for the correction.

Line 19: Not grammatically correct. Could be something like "The high frame rates allowed us to obtain better resolved time derivatives of velocity..."

We will change to: " The high frame rates enabled us to calculate time derivatives to obtain velocity and acceleration fields."

Line 29: citation?

We will cite: Pudasaini and Hutter (2007) (ISBN 978-3-540-32687-8) and Schweizer et al. (2021) (ISBN 978-0-12-817129-5).

Line 44: This definition isn't accurate as the weak layer is always in contact with the slab since, as the authors state in the discussion, the slab is never in free fall.

Thanks, for pointing out. We will rewrite to:" Quantities of particular interest during self-sustained crack propagation are the speed of the propagating crack, the touchdown distance, which is the length from the crack tip to the trailing point where the slab rests on the crushed weak layer, and the specific fracture energy of the weak layer (e.g., Schweizer et al., 2011; van Herwijnen et al., 2016b; van Herwijnen et al., 2010)"

Line 48: why not abbreviate as PST starting here?

We will follow your suggestion and abbreviate at this point.

Line 57: no space
Line 73: These high fracture energies are comparable to solid ice (0.3-2 J m^-2) as pointed out by Dave McClung in a review of van Herwijnen et al. (2016) and Reuter et al (2019). That suggests that either: 1) there's something wrong with the E and wf measurements or 2) there's quite a bit of dissipation of that energy.
Rosendahl and Weissgraeber (2020, p 126) have a nice discussion about this and suggest that the compressive fracture toughness (which can be related to the sp. fracture energy with E) for snow should be significantly higher than the tensile fracture toughness for ice because of dissipative processes involved in the crushing of the weak layer.
I don't expect the authors to provide a definitive answer, but more context on these specific fracture energies is needed.

We are aware that this point was raised previously. We completely agree with the arguments of Rosendahl and Weissgraeber (2020). Tensile failure and compressive failure are very different. That holds for strength of materials approaches (tensile strength and compressive strength are different properties) as well as for fracture mechanic approaches (specific fracture energy in tensile is not the same as specific fracture energy in compression). We therefore

think that the remark, initially raised by Dave McClung, is comparing two different properties, and is therefore not adequate to claim the contradiction of too high specific fracture energies for snow. We will include a discussion of this "contradiction" in the revised manuscript.

Line 75: should be "derivation of". I suggest an English language service and won't make further grammatical corrections.

Thanks for the suggestion.

We will define time (t) one line above

Line 170: Might want to mention that accounting for material properties of the weak layer is the major difference between RW model and the Heierli model, which assumes a slab in free fall.

We will mention the difference by writing:
*"Their model [RW method] consists of a Timoshenko beam sitting on a weak layer represented by smeared springs, in contrast to the model of Heierli et al. (2008a), where the weak layer is rigid."*

Line 223: The slab is resting on the weak layer prior to failure so it's always in contact with the weak layer and whether or not the weak layer is crushed is a qualitative description at best. And as the authors mention in the discussion, the slab is never in free fall.
What the authors are describing is the section of the slab that is experiencing (positive) slope normal movement.

We agree, "making contact again" is not correct. We will rewrite to:
*"As the crack propagates through the PST beam, the slab subsides before it comes to rest on the crushed weak layer."*

Line 223: Looks like there are some edge effects to address, i.e. velocity is highest closest the far edge of the beam.

Yes, the downward velocity of subsets which are close to the far end of the beam increases. We address this edge effects later in the discussion around line 395.

Figure 7: This key is not intuitive and make the melt freeze layer look very thick at first glance. Maybe just have the symbols with arrows pointing towards each layer?

Thanks for the suggestion. We think the many arrows will make the graph messy. Instead we will modify the graph by plotting the grain type legend more separated.

[Figure]

Table 1: Where is the slope angle? That's important.

As stated in line 91, all PSTs were performed in the flat. We will add this to the caption of Table 1.

Line 264: This is interesting. I think stress intensification from the far edge is playing a big role here and the secondary crack may be an artifact of that. Could the second crack going in the opposite direction be seen visually, or only using DIC? It's pertinent because there aren't many field PST observations (without high speed measurements) where this occurs.

Indeed, the far edge plays an important role. The secondary crack is initiated by the impact of the stronger negative acceleration the slab experiences at the far end of the beam when sitting down on the crushed weak layer.  We did not observe the secondary weak layer cracking nor the slab fractures in the field or in the videos. In a revised version we will highlight this by writing:
*"While in the field we classified PST3 as END, the displacement and strain data clearly show that the crack propagation dynamics were more intricate, and a combination of END, SF and ARR. This unexpected result was not recognized in the field."*

Line 348: 26 This is an overstatement. This is not the first time strain fields have been measured in snow at high speed & high resolution, e.g. Reweiger and Schweizer 2013.

We agree, to be more precise, and to distinguish from the work done by Reiweger and Schweizer 2013, we will rewrite to: *"For the first time, we were able to measure strain fields in PSTs, showing strain concentrations in the area of the weak layer (Figure 8) as well as in the slab in experiments with slab fractures."*

Line 355: Ah, this should be stated in the results.

We will also mention this in the Results section (see above).

Line 359: Again, difficult to rule out artifacts from edge effect

See reply on comment line 264. We also suggest that the secondary crack is caused by an edge effect. However, we think that further discussion about edge effects would distract the reader at this point. That's why we refer to Appendix A.

Line 373: Not sure the grain scale measurements would be helpful as especially with something like large surface hoar crystals, you'd see many different fracture modes as the grain get blown apart during the collapsing and shearing of the slab.

We agree and already discuss this in lines 367-374.

Line 409: This whole idea of steady-state crack propagation may be a red herring. It's never observed in controlled experiments. And we know that spatial variability is the rule with snow in the mountains, which is why we see wild looking river markings on crown faces, even in new snow (e.g. Fig 4 in Bair et al 2016), suggesting cracks traveling at different speeds as they encounter snow with different properties.
Bair, E.H., Gaume, J. and van Herwijnen, A. (2016). The role of collapse in avalanche release: review and implications for practitioners and future research, Proceedings of the 2016 International Snow Science Workshop, Breckenridge, CO USA.

Our experiments come close to controlled experiments. Laterally very homogenous snowpack, flat field (cf. Fig. 7), and the beam length of PST #3 was larger than the touchdown distance. We therefore think that it is worth mentioning that a theoretically predicted steady-state for such conditions was not (yet) observed. That spatial variability most likely causes the crack speed to adapt along its path is thus not called into question.
We will somewhat reword this sentence. Instead of writing 'we were not able to observe', which suggests that we failed to observe something we assume is there, we will write 'we did not observe'.

Line 412: I believe these collapse wave speed measurements, but they are slower than speeds measured from real avalanches (Hamre et al. 2014), or slope scale simulations (Game et al, 2019). Thus some discussion about measuring collapse wave speeds from PSTs and how they relate to avalanches is warranted. I assume the PSTs were conducted on low angle slopes, but slope angle measurements are not provided in Table 1.
Gaume, J., van Herwijnen, A., Gast, T., Teran, J., & Jiang, C. (2019). Investigating the release and flow of snow avalanches at the slope-scale using a unified model based on the material point method. Cold Regions Science and Technology, 168, 102847. https://doi.org/https://doi.org/10.1016/ j.coldregions.2019.102847
Hamre, D., Simenhois, R., & Birkeland, K. (2014). Fracture speeds of triggered avalanches. Presented at the International Snow Science Workshop, Banff.

Indeed, our experiments were performed in the flat. We agree that different circumstances, e.g. slope angle > 30°, may lead to different crack propagation modes and therefore to much higher crack speeds and will address this in the Introduction section of the revised manuscript.

Line 420: And because of the edge effects, that maximum is probably greater than what you'd see in an avalanche or whumpf in the field with the same slab/weak layer.

We agree, but we do not think this assumption is relevant enough to be mentioned in this context.

Line 420: Is there a practical takeaway here? Are there implications for practitioners using PSTs?

The long touchdown lengths, in addition to edge effects at both beam ends, make it impossible to assess the propensity for self-sustained crack propagation with normal-sized PSTs. We will mention this in a revised version with:

"As a practical implication, our results show the need to rethink the predictive power of normal-sized PST experiments. That the measured touchdown distance is longer than a typical PST beam length once more emphasizes that normal-sized PSTs cannot be used to assess the propensity for self-sustained crack propagation."

Line 494: Not compliant. Either make the videos freely available or explain why they are not see: https://www.the-cryosphere.net/policies/data_policy.html#data_availability

We will refer to WSL's data repository [www.envidat.ch](www.envidat.ch) in the revised manuscript and make the data available on acceptance of the manuscript.

---

## Author Response (AR1)

Dear Dr. Rosendahl, dear Dr. Bair,

thank you very much for commenting and providing helpful suggestions on the manuscript. Below we have pasted your comments in blue. In black, you find our point-by-point responses on how we changed the manuscript due to your remarks. The line numbers in our answers refer to the revised manuscript (whereas we did not alter the line numbers in your comments). The first seven pages contain responses to remarks by P. Rosendahl. Starting with page eight, we address the remarks of E. Bair.

**Review 1:**

The manuscript presents a methodology for full-field measurements of snowpack displacements using digital image correlation. The work opens numerous possibilities for the extraction of snowpack properties. Among these, the authors discuss ways to obtain an effective homogenized elastic modulus of the slab, the weak layer fracture toughness and the speed of cracks running in the weak layer. The study focuses on the comparison of different methodologies using three representative examples. The paper makes a significant contribution towards the understanding and characterization of fracture mechanical processes that lead to slab avalanche release. However, I have a serious concern regarding the derivation of the weak-layer fracture energy from SMP signals, denoted $w_f^{BR}$.
The manuscript cites Reuter and Schweizer (2018) [doi: 10.1029/2018GL078069] for a description of the approach. In this work, however, I find no explanation of the methodology. Instead, I am referred to Reuter et al. (2018) [doi: 10.16904/envidat.40], which, again, does not clarify the procedure. In the accompanying README file I am referred to publication: Reuter et al. (2015) [doi: 10.5194/tc-9-837-2015], Eq. (4). Here, the fracture energy is obtained from the integration of the SMP force signal over certain windows and subsequent selection of the minimum value within the weak layer:

$$w_f = \min_{\mathrm{WL}} \int_{-\frac{w}{2}}^{+\frac{w}{2}} F \mathrm{d}z, \tag{1}$$

where w is the windows size and F the penetration resistance. From the publication I understand that w is of dimension length and F of dimension force (e.g., Reuter et al. (2015) [doi: 10.5194/tc-9-837-2015], Figure 3). This yields units of Nm (energy) for $w_f$ when it should be N/m = J/m2 (energy per unit area). The following thought experiment raises another concern about the above equation (1). Imagine we probe the same weak layer (with the same fracture energy) with an SMP of twice or half the original diameter. The former should yield a larger resistance F, the latter a smaller one. Evaluating all three signals (original, double, and half diameter) with the same window size will yield three different fracture toughnesses of the same weak layer. Which one is correct? Moreover, Reuter et al. (2015) [doi: 10.5194/tc-9-837-2015] refer to Reuter et al. (2013) [url: http://arc.lib.montana.edu/snow-science/objects/ISSW13_paper_O2-02.pdf] regarding the validation of the methodology surrounding the above equation (1). Here, the accuracy of the method is checked using an approach similar to the VH method used in the present work. However, in the present manuscript, the VH method is deemed unfit for the derivation of $w_f$, for instance because of its inability to model the measured strain energy (Figure 3a) or its inability to correctly account for the slab's Young modulus (Figure 10a). I encourage the authors to comment on this contradiction because I cannot understand the details of the procedure used by Reuter et al. (2013) since no equations are given. Concluding my concerns surrounding $w_f^{BR}$, I specifically ask for clarification of the following:

1. Please explicitly explain (in the manuscript) how the fracture toughness $w_f^{BR}$ is derived from SMP signals including corresponding equations and dimensions.

We regret that information was missing to understand the calculation of fracture energy derived from SMP measurements. We now included the necessary information in the Methods section of the revised manuscript (lines 192-204):

*"As a third approach, we used SMP measurements. Effective elastic modulus $E_{sl}^{BR}$ was derived from SMP data as described by Reuter and Schweizer (2018), using the signal interpretation method suggested by Löwe and van Herwijnen (2012). Reuter et al. (2013) suggested a parametrization of the specific fracture energy based on the penetration resistance F(z). Using a moving window (size: w = 2.5 mm) to integrate F(z), they then defined the specific fracture energy as the minimum of the integral within the weak layer:*

$$w_f^{SMP} = \frac{1}{A} \min_{WL} \int_{-w/2}^{w/2} F\, dz \, , \qquad\qquad 2$$

*Where A is a fitting parameter. The integration has units of energy (J) and relates to the work required to destroy the snow structure along the integration path. Specific fracture energy, however, has unit energy per area. Therefore, it is necessary to divide by an effective area, the fitting parameter A. While the effective area is unknown, it is likely larger than the cross section of the tip diameter (Johnson, 2003), and depends on snow structure (van Herwijnen, 2013). We therefore followed Reuter et al. (2019), and introduced a fitting parameter A to implicitly account for the unknown effective area. The fitting parameter was derived using a linear regression to PTV derived specific fracture energies (Figure 6 in Reuter et al. (2019)), resulting in A = 3.4 × 10-4 $m^2$, which relates to a plausible effective cone area of 3.4 $cm^2$ (radius ≈ 1 cm)."*

2. Please comment on the units of $w^{BR}_f$ and Eq. (1) above.

The unit discrepancy originated from the absence of the fitting factor *A* (see above, equation 2 in the reply to your comment 1). Its physical meaning and derivation is now explicitly stated in the revised version of the manuscript (see reply above).

3. Please comment on the issue different probe diameters regarding Eq. (1) above.

With the fitting parameter A, accounting for the effective area, the equation to derive the fracture energy from SMP data now accounts for the probe size (see reply above).

4. Please comment on whether I correctly understood the validation of Eq. (1) in Reuter et al. (2013) and the consequential contradiction.

Given our reply above we hope that the issue is now clarified.
With the fitting parameter we introduced in Equation 2, it becomes clear now that the comparison to PTV-derived values in Reuter et al. (2013) and Reuter et al. (2019) served to parametrize the specific fracture energy on SMP signals. Reuter et al. (2013) and Reuter et al. (2019) showed a comparison of SMP- and PTV-derived values of the specific fracture energy. Rather than validating the accuracy of the SMP method, they discussed differences between different measurement methods. Currently lacking an alternative method for calibration, we used their PTV data to determine the factor *A* in equation 2.
Based on your comment 12 below, we applied corrections to the VH method (Fig. 3a), which improved the fit of the mechanical energy. Nevertheless, the SMP-derived values are based on a parameterization derived from a linear regression with PTV-derived values. Once enough data are available, we will derive a parameterization using DIC data, or μCT data, or other future techniques, which may possibly provide more accurate $w_f$ data. We deem it

valuable to provide the SMP-derived estimates and to compare them to more elaborate methods. Only then we know how the method performs and we can possibly calibrate the SMP approach to the best method in the future. For lack of alternative, the SMP currently remains the only efficient experimental method to determine snow mechanical properties, such as the specific fracture energy, at many locations in the field.

These points are now discussed explicitly in the in the revised version of the manuscript (lines 441-445):

These points should be clarified beyond doubt. If a comprehensive discussion of the methodology goes beyond the scope of the present work, I suggest omitting the SMP methodology for now. After all, its connection to high-resolution and high-speed photography is weak.

We agree that the main message of the paper is the potential of high-resolution DIC measurements. As the SMP is presently the only measurement method to provide quantitative snow stratigraphy data related to mechanical properties, we decided to keep the SMP results. Moreover, we deem comparisons with existing methods good practice. However, we substantially reduced the importance of the SMP-derived values by only mentioning those in the text, and not in the figures anymore, and we explained the derivation of the fracture energy in greater detail, as mentioned above.

Aside from the above crucial points, I only have one other major remark:
5. Since you extract the external potential Vp directly from measured full-field data, Clapeyron's Theorem allows for direct identification of the total potential Vtot = Vm + Vp = Vp/2 and, hence, direct computation of the fracture energy wf = dVtot/dr = dVp/(2dr). No fitting to an analytical expression, only some form of signal processing of the experimental data shown in Figure 3a is required to compute the derivative.

Thank you for your suggestion. As Heierli's formulation of the mechanical energy did not represent the measured data very well, we followed your suggestion by fitting an arbitrary function to our data. We only had two constraints for the function: 1) it should have a value of 0 for $r = 0$, and 2) the function should be monotonically decreasing with $r$. We used a simple power law function of the form $f(r) = -ar^b$ (FU) and refined the fitting window to 15 cm $< r_{max} < r_c$.
In the end, we assessed the quality of the VH and FU fit with the root mean squared error and found that the simple power law function FU represented the measured data better ($RMSE^{FU} = 0.007$, $RMSE^{VH} = 0.013$).
We introduced the power law fit into the revised manuscript in
  - the Methods section (lines 171-174)
  - Figure 3a
  - Table 2
  - Figure 10b
  - The Results section (line 324)
  - The Discussion section (around line 439-441)

Finally, I ask the authors to consider the following minor remarks:

6. The abstract devotes considerable attention to historical developments (lines 10–15) but does not include key findings of the manuscript. I suggest moving the historical perspective to the introduction and add key results such as determined crack speeds and fracture toughnesses – including the respective most suitable techniques for their identification.

We agree that the historical perspective was rather prominent and revised the Abstract.

7.  (line 31) How does process of coalescence of subcritical failures work?

Coalescence of subcritical failures is part of our conceptual understanding of natural avalanche release (Schweizer et al., 2016). The formation of subcritical failures occurs at the microscale (scale of snow crystals and bonds, <<1 mm). At this scale two competing processes occur simultaneously (Capelli et al., 2018a): 1. Weak layer damage (meaning the breaking of bonds), and 2. Weak layer strengthening/sintering (meaning creation and strengthening of bonds). When the damage process dominates, more bonds break and a localized failure may develop, i.e. subcritical failures coalesce.
This damage process, aka failure events (bond breaking), manifests itself by acoustic emissions (Capelli et al., 2018b).

8.  (line 47) Please motivate and discuss why and how crack speed is important.

We motivated the importance of crack speed by pointing out that crack speed is linked to crack arrest phenomena (lines 44-48).

9.  (line 60) The touchdown length is not a material constant but depends, for instance, on the slab's bending stiffness and its density $\rho$. In order to give context to the listed absolute values, I suggest adding additional information.

Indeed, the touchdown length is not a material property. We now provide the range of slab densities and slab thickness reported in Bair et al. (2014) in line 61.

10.     (line 68) The statement is a bit misleading. The fracture energy itself is an independent fundamental material property and independent of other fundamental properties such as the elastic modulus. I assume what is meant is the following: because the method employs a certain model to compute wf , and the model requires E as and input, the back calculation will change if E changes?

We agree with the reviewer that the statement was unclear. We therefore reworded the sentence (lines 78-79).

11.  (line 130) Can you provide examples of used reference lengths?

In all tests we acquired an image with a 2 m reference length fixed on the PST side wall. We explicitly mention this now in line 139.

12.  (lines 157–162) The equation in line 162 only holds if Vm and, hence, also Vp in line 157 are defined per unit width. Please explicitly state (in an equation) how Vp is determined. Is layering considered?

We thank the reviewer for pointing out this mistake. Indeed, we did not define $V_p$ per unit width, resulting in wrong estimates of the elastic modulus and specific fracture energy of the VH method. We corrected this error and made the necessary changes throughout the revised manuscript.

13. (line 175) Equation number missing.

We added the equation number.

14. (lines 413–414) Can you discuss possible reasons for this discrepancy? How does weak-layer rigidity or compliance affect crack speed?

In Heierli (2005), speed is proportional to the $4^{th}$ root of the bending stiffness of the slab, which itself is directly proportional to the elastic modulus. Computing the speed with the VH and RW elastic modulus gives values of $c^{VH} = 25$ m s$^{-1}$ and $c^{RW} = 35$ m s$^{-1}$.
The model of Heierli assumes free fall motion of the slab during weak layer collapse, and therefore does not consider weak layer properties. Accounting for weak layer rigidity would therefore likely reduce the speed estimates.

15. (line 424) Clapeyron's Theorem is a fundamental law of mechanics and should not be brought in context with the limitations of certain models. Instead, I suggest to explicitly repeat arguments for weaknesses of the VH method that were given around line 301.

We do no longer mention Clapeyron´s Theorem in the revised manuscript and modified the argumentation (lines 434-437).

Figures and Tables:
16. All images seem to have a low resolution and show compression artifacts. Is this a draft issue?

We improved the resolution of all figures.

17. (Figure 1) Images are very small.

We enlarged the three images with the limitation to fit everything in a single line. Since the intention of the figures is to present a schematic workflow of the processing, it was more important for us to keep a single line instead of showing the steps in detail.

18. (Figure 2) Red text on gray picture is hard to read.

We agree and improved Figure 2 by removing text transparency and changing text color.

19. (Figure 10b) Why does $w^{RW}_f$ decrease with rsaw? I would expect the contrary. Is a constant Young's modulus chosen for each data point or does is change alongside rsaw? I would suggest to use the "converged" effective modulus from Figure 10a (for both the VH and the RW methods) to calculate the fracture energies in 10b.

We do not see why the contrary should be expected. With a "perfect measurement" and a "perfect model" we would not expect any trend. Since this is, however, never the case, we investigated how strong the elastic modulus (moduli) varies when changing the fit interval ($r < r_{max}$, VH method) or when taking another measured displacement field ($r = r_{max}$, RW method). In a further step the weak layer fracture energies $w_f$ are derived from the elastic properties as:

$$w_f^{\text{VH}} = -\left.\frac{d}{dr}V_{\text{m}}\right|_{r=r_c} \; ; \; w_f^{\text{RW}} = G_{\text{I}} + G_{\text{II}} = \frac{E_{\text{wl}}^{\text{RW}}}{2t}w_{\text{RW}}(r=r_c)^2 + \frac{E_{\text{wl}}^{\text{RW}}}{2t(v_{\text{wl}}-1)}u_{\text{RW}}(r=r_c)^2$$

Therefore, the variation of elastic properties propagates into $w_f^{\text{VH}}$ and $w_f^{\text{RW}}$. To illustrate that, Figure 10b shows how the variation of the elastic moduli affects the derived values of $w_f$.

Using a "converged modulus" would result in one specific fracture energy for each method. These energies are basically already given as the data points with largest $r_{\text{max}}$ in Figure 10b.

To avoid misunderstandings, we now mention this explicitly (lines 325-328).

20. (Table 2) Again, please check the units of wBR f .

This should now be clarified given our reply to your comment 2 above.

21. (Table 3) The mean of $c_{\text{corr}}$ suffers from (potential) inaccuracies towards the boundaries. Does it make sense to introduce a fourth column where the mean is evaluated on a more reasonable x-domain?

We are unsure whether you suggest to introduce a fourth column because you may oversee the last line in Table 3, in which the mean crack speed away from beam edges
(1 m < $x$ < 2 m) is shown.
Or, do you question if the last line in Table 3 is meaningful at all?
In that case we think it is useful to separate two regions and state mean crack speeds within. Our discussion also considers drivers of crack speed changes within these two regions and the regions are:
1.      Close to beam ends, where strong edge effects are to be expected.
2.      The middle part of the beam, where edge effects are less pronounced but still present as long as the crack is not in a steady state propagation.

**References**

Bair, E. H., Simenhois, R., van Herwijnen, A., and Birkeland, K.: The influence of edge effects on crack propagation in snow stability tests, The Cryosphere, 8, 1407-1418, 10.5194/tc-8-1407-2014, 2014.

Capelli, A., Reiweger, I., Lehmann, P., and Schweizer, J.: Fiber bundle model with time-dependent healing mechanisms to simulate progressive failure of snow, Physical Review E, 98, 023002, 10.1103/PhysRevE.98.023002, 2018a.

Capelli, A., Reiweger, I., and Schweizer, J.: Acoustic emissions signatures prior to snow failure, Journal of Glaciology, 64, 543-554, 10.1017/jog.2018.43, 2018b.

Heierli, J.: Solitary fracture waves in metastable snow stratifications, Journal of Geophysical Research, 110, F02008, doi: 02010.01029/02004JF000178, 2005.

Johnson, J. B.: A statistical micromechanical theory of cone penetration in granular materials, U.S. Army Corps of Engineers, Engineer Research and Development Center, Hanover NH, U.S.A., ERDC/CRREL Technical Report, ERDC/CRREL-TR-03-3, 32, 2003.

Löwe, H., and van Herwijnen, A.: A Poisson shot noise model for micro-penetration of snow, Cold Regions Science and Technology, 70, 62-70, 10.1016/j.coldregions.2011.09.001, 2012.

Reuter, B., and Schweizer, J.: Describing snow instability by failure initiation, crack propagation, and slab tensile support, Geophysical Research Letters, 45, 7019-7027, 10.1029/2018GL078069, 2018.

Reuter, B., Proksch, M., Löwe, H., van Herwijnen, A., and Schweizer, J.: Comparing measurements of snow mechanical properties relevant for slab avalanche release, Journal of Glaciology, 65, 55-67, 10.1017/jog.2018.93, 2019.

Schweizer, J., Reuter, B., van Herwijnen, A., and Gaume, J.: Avalanche release 101, Proceedings ISSW 2016. International Snow Science Workshop, Breckenridge CO, U.S.A., 3-7 October 2016, 1-11, 2016.

van Herwijnen, A.: Experimental analysis of snow micropenetrometer (SMP) cone penetration in homogeneous snow layers, Canadian Geotechnical Journal, 50, 1044-1054, 10.1139/cgj-2012-0336, 2013.

**Review 2:**

1   Line 10: "is" b.c. information is singular.

Thanks, for the correction. In the course of the English correction (see comment "Line 75"), this sentence was deleted.

2   Line 19: Not grammatically correct. Could be something like "The high frame rates allowed us to obtain better resolved time derivatives of velocity..."

We changed to: "*The high frame rates enabled us to calculate time derivatives to obtain velocity and acceleration fields.*" (line 17).

3   Line 29: citation?

We now cite: Pudasaini and Hutter (2007) (ISBN 978-3-540-32687-8) and Schweizer et al. (2021) (ISBN 978-0-12-817129-5) (line 27).

4   Line 44: This definition isn't accurate as the weak layer is always in contact with the slab since, as the authors state in the discussion, the slab is never in free fall.

Thanks, for pointing out. We revised the description of the touchdown distance (lines 40-42)

5   Line 48: why not abbreviate as PST starting here?

We followed your suggestion and abbreviate at this point (line 50).

6   Line 57: no space
7   Line 73: These high fracture energies are comparable to solid ice (0.3-2 J m^-2) as pointed out by Dave McClung in a review of van Herwijnen et al. (2016) and Reuter et al (2019). That suggests that either: 1) there's something wrong with the E and wf measurements or 2) there's quite a bit of dissipation of that energy.
Rosendahl and Weissgraeber (2020, p 126) have a nice discussion about this and suggest that the compressive fracture toughness (which can be related to the sp. fracture energy with E) for snow should be significantly higher than the tensile fracture toughness for ice because of dissipative processes involved in the crushing of the weak layer.
I don't expect the authors to provide a definitive answer, but more context on these specific fracture energies is needed.

We are aware that this point was raised previously. We completely agree with the arguments of Rosendahl and Weissgraeber (2020). Tensile failure and compressive failure are very different. That holds for strength of materials approaches (tensile strength and compressive strength are different properties) as well as for fracture mechanics approaches (specific fracture energy in tension is not the same as specific fracture energy in compression). We therefore think that the remark, initially raised by Dave McClung, is comparing two different properties, and is therefore not adequate to claim the contradiction of too high specific fracture energies for snow. We included a discussion of this "contradiction" in the Introduction section (lines 64-72).

8   Line 75: should be "derivation of". I suggest an English language service and won't make further grammatical corrections.

Thanks for the suggestion. We paid attention to improving language while revising the manuscript.

9   Line 123: define t as time. In seconds I assume?

We defined time (t) one line above (line 131).

10  Line 170: Might want to mention that accounting for material properties of the weak layer is the major difference between RW model and the Heierli model, which assumes a slab in free fall.

We now mention the main difference between the models (lines 181-183).

11  Line 223: The slab is resting on the weak layer prior to failure so it's always in contact with the weak layer and whether or not the weak layer is crushed is a qualitative description at best. And as the authors mention in the discussion, the slab is never in free fall.
What the authors are describing is the section of the slab that is experiencing (positive) slope normal movement.

We agree, "making contact again" is not correct. We rewrote to (line 284):*"As the crack propagates through the PST column, the slab subsides before it comes to rest on the crushed weak layer."*

12  Line 223: Looks like there are some edge effects to address, i.e. velocity is highest closest the far edge of the beam.

Yes, the downward maximum velocity of subsets that are close to the far end of the beam increases. We address these edge effects later in the discussion around lines 404-409.

13  Figure 7: This key is not intuitive and make the melt freeze layer look very thick at first glance. Maybe just have the symbols with arrows pointing towards each layer?

Thanks for the suggestion. We think the many arrows would have made the graph messy. Instead, we modified the graph by plotting the grain type legend on the side.

[Figure]

14  Table 1: Where is the slope angle? That's important.

As stated in line 96, all PSTs were performed in the flat. We now added this information to the caption of Table 1.

15  Line 264: This is interesting. I think stress intensification from the far edge is playing a big role here and the secondary crack may be an artifact of that. Could the second crack going in the opposite direction be seen visually, or only using DIC? It's pertinent because there aren't many field PST observations (without high speed measurements) where this occurs.

Indeed, the far edge plays an important role. The secondary crack is initiated by the impact of the stronger negative acceleration the slab experiences at the far end of the beam when sitting down on the crushed weak layer. We did not observe the secondary weak layer cracking nor the slab fractures in the field or in the videos. We now highlight this: "*While in the field we classified PST3 as END, the displacement and strain data clearly show that the crack propagation dynamics were more intricate, and a combination of END, SF and ARR. This unexpected result was not recognized in the field.*" (lines 280-282).

16  Line 348: 26 This is an overstatement. This is not the first time strain fields have been measured in snow at high speed & high resolution, e.g. Reweiger and Schweizer 2013.

We agree, to be more precise, and to distinguish from the work done by Reiweger and Schweizer 2013, we rewrote to: *"For the first time, we were able to measure strain fields in a PST, showing strain concentrations in the area of the weak layer (Figure 8) as well as in the slab in experiments with slab fractures."* (lines 361-363).

17  Line 355: Ah, this should be stated in the results.

We now also mention this in the Results section (see comment 15).

18  Line 359: Again, difficult to rule out artifacts from edge effect

See reply on comment 15. We also suggest that the secondary crack is caused by an edge effect. However, we think that further discussion about edge effects would distract the reader at this point. That's why we refer to Appendix A.

> 19 Line 373: Not sure the grain scale measurements would be helpful as especially with something like large surface hoar crystals, you'd see many different fracture modes as the grain get blown apart during the collapsing and shearing of the slab.

We agree and already discuss this point in lines 379-384.

> 20 Line 409: This whole idea of steady-state crack propagation may be a red herring. It's never observed in controlled experiments. And we know that spatial variability is the rule with snow in the mountains, which is why we see wild looking river markings on crown faces, even in new snow (e.g. Fig 4 in Bair et al 2016), suggesting cracks traveling at different speeds as they encounter snow with different properties.
> Bair, E.H., Gaume, J. and van Herwijnen, A. (2016). The role of collapse in avalanche release: review and implications for practitioners and future research, Proceedings of the 2016 International Snow Science Workshop, Breckenridge, CO USA.

Our experiments come close to controlled experiments. Laterally very homogeneous snowpack, flat field (cf. Fig. 7), and the beam length of PST #3 was larger than the touchdown distance. We therefore think that it is worth mentioning that a theoretically predicted steady-state for such conditions was not (yet) observed. That spatial variability most likely causes the crack speed to adapt along its path is thus not called into question. We reworded this sentence. Instead of writing 'we were not able to observe', which suggests that we failed to observe something we assume is there, we now write 'we did not observe' (line 419). Further, we note that the existence of such a "steady-state crack propagation" could be clarified by performing very long PST experiments (line 412).

> 21 Line 412: I believe these collapse wave speed measurements, but they are slower than speeds measured from real avalanches (Hamre et al. 2014), or slope scale simulations (Gaume et al, 2019). Thus some discussion about measuring collapse wave speeds from PSTs and how they relate to avalanches is warranted. I assume the PSTs were conducted on low angle slopes, but slope angle measurements are not provided in Table 1.
> Gaume, J., van Herwijnen, A., Gast, T., Teran, J., & Jiang, C. (2019). Investigating the release and flow of snow avalanches at the slope-scale using a unified model based on the material point method. Cold Regions Science and Technology, 168, 102847. https://doi.org/https://doi.org/10.1016/ j.coldregions.2019.102847
> Hamre, D., Simenhois, R., & Birkeland, K. (2014). Fracture speeds of triggered avalanches. Presented at the International Snow Science Workshop, Banff.

Indeed, our experiments were performed in the flat. We agree that different circumstances, e.g. slope angle > 30°, may lead to different crack propagation modes and therefore to much higher crack speeds.

> 22 Line 420: And because of the edge effects, that maximum is probably greater than what you'd see in an avalanche or whumpf in the field with the same slab/weak layer.

We agree, but we do not think this assumption is relevant enough to be mentioned in this context.

> 23 Line 420: Is there a practical takeaway here? Are there implications for practitioners using PSTs?

The long touchdown lengths, in addition to edge effects at both beam ends, make it impossible to assess the propensity for self-sustained crack propagation with normal-sized PSTs. We now mention this in the revised manuscript (lines 429-431).

> 24 Line 494: Not compliant. Either make the videos freely available or explain why they are not see: https://www.the-cryosphere.net/policies/data_policy.html#data_availability

In the revised manuscript, we now refer to WSL's data repository www.envidat.ch and make the data available on acceptance of the manuscript.

---

## Author Response (AR2)

Dear Dr. Bair,

Thank you for taking time again to review our paper. Below we have pasted your comment in blue and our response as well as how we changed the manuscript in black.

The authors failed to fully respond to my review. On p.8 of the combined response, they address all the minor critiques but do not respond to the major issue. To rehash, in the first review, I asked the authors to better motivate their study:

"My major critique is that no attempt is made to link these measurements to slope scale avalanches or practical use, which should be overarching goals. Since its inception, the PST has been used to study fracture in snow, however we know that, as with any small-scale stability test that involves isolated blocks of snow, it is contrived and not fully representative of the avalanche process. Recent work (e.g. Gaume et al., 2019) suggests that the PST can effectively represent collapse waves in low angle terrain, but that the exaggerated bending is not representative of slope scale failure. For example, slab fracture in the PST begins at the top of the snowpack, while the simulated crowns in Gaume et al. (2019) open from the bottom. Crack speeds measured in avalanches (Hamre et al., 2014) are several times faster than 21-30 m/sec values measured in the PSTs here. Thus, I suggest further discussion on the motivation and utility of these high speed PST measurements towards understanding the avalanche process. Why are we still doing PSTs and carefully studying them? "

This omission strikes me as an oversight, but it still should have been addressed

As you, we are keen to know how indicative PST's are for assessing snow slope stability. However, this manuscript is essentially a methods paper where we present and evaluate the DIC method to study crack propagation in snow with the PST with unprecedented detail.

The PST is a well-established test to assess the onset of crack propagation. The improved temporal resolution in our experiments allows us for the first time to investigate the dynamics of crack propagation in a PST. Whether the PST is a suitable test to investigate dynamic crack propagation, and how representative it is for the fracture process in avalanches is a legitimate question, but goes far beyond the scope of our work. The three flat field PSTs presented in the manuscript represent three typical outcomes of a PST: slab fracture, crack arrest and full propagation. Based on these three examples, we cannot make any conclusions about the relevance of these PSTs for the avalanche release process in general.

Of course, we are aware of the publications and the very recent modelling approaches. All the recent and ongoing research efforts will undoubtedly contribute to answer the question you rise in the near future.

To take up the issue you raised we amended the Introduction section lines 40 – 47:

*"The Propagation Saw Test (PST), a fracture mechanical field experiment for snow (Sigrist et al., 2006;Gauthier and Jamieson, 2006), can resolve processes at the snowpack scale. It was intensely used to study the onset of crack propagation (e.g., Birkeland et al., 2019;van Herwijnen et al., 2016). If the PST is a proper test to study self-sustained crack propagation and thus relates to slope scale processes is an open question. To the best of our knowledge,*

*no study shows that the PST geometry (isolated beam) has an influence on self-sustained crack propagation and recent findings suggest that crack propagation speeds measured during PST experiments may be indicative for slope scale processes (Bergfeld et al., 2020). However, quantities characterizing self-sustained crack propagation may depend on PST length, snowpack characteristics and slope angle, as these parameters influence crack propagation (Gaume et al., 2019)."*

And in line 61 – 63, where we also mention the higher crack speeds reported by Gaume et al. (2019) and Hamre et al. (2014):

*"However, much higher crack speeds were estimated as well (Hamre et al., 2014;Gaume et al., 2019;Trottet et al., 2021). This highlights again, that reported PSTs do not cover the full parameter space, especially in terms of PST length."*

Bergfeld, B., van Herwijnen, A., Bobillier, G., and Schweizer, J.: Measuring slope-scale crack propagation in weak snowpack layers, EGU General Assembly Conference 2020,  May 1 2020, 2020.

Birkeland, K. W., van Herwijnen, A., Reuter, B., and Bergfeld, B.: Temporal changes in the mechanical properties of snow related to crack propagation after loading, Cold Regions Science and Technology, 159, 142-152, 10.1016/j.coldregions.2018.11.007, 2019.

Gaume, J., van Herwijnen, A., Gast, T., Teran, J., and Jiang, C.: Investigating the release and flow of snow avalanches at the slope-scale using a unified model based on the material point method, Cold Regions Science and Technology, 168, 102847, 10.1016/j.coldregions.2019.102847, 2019.

Gauthier, D., and Jamieson, J. B.: Evaluating a prototype field test for weak layer fracture and failure propagation, Proceedings ISSW 2006. International Snow Science Workshop, Telluride CO, U.S.A., 1-6 October 2006, 2006, 107-116, 2006.

Hamre, D., Simenhois, R., and Birkeland, K.: Fracture speeds of triggered avalanches, Proceedings ISSW 2014. International Snow Science Workshop, Banff, Alberta, Canada, 29 September - 3 October 2014, 2014, 174-178,

Sigrist, C., Schweizer, J., Schindler, H. J., and Dual, J.: Measurement of fracture mechanical properties of snow and application to dry snow slab avalanche release, Geophysical Research Abstracts, 8, 08760, 2006.

Trottet, B., Simenhois, R., Bobillier, G., van Herwijnen, A., Jiang, C., and Gaume, J.: From sub-Rayleigh to intersonic crack propagation in snow slab avalanche release, EGU General Assembly 2021, online, 19–30 Apr 2021, 2021.

van Herwijnen, A., Gaume, J., Bair, E. H., Reuter, B., Birkeland, K. W., and Schweizer, J.: Estimating the effective elastic modulus and specific fracture energy of snowpack layers from field experiments, Journal of Glaciology, 62, 997-1007, 10.1017/jog.2016.90, 2016.